

# Ozone trend profiles in the stratosphere: combining ground-based data over Central Europe to consider uncertainties

Leonie Bernet[1,2], Thomas von Clarmann[3], Sophie Godin-Beekmann[4], Gérard Ancellet[4], Eliane Maillard Barras[5], René Stübi[5], Wolfgang Steinbrecht[6], Niklaus Kämpfer[1,2], and Klemens Hocke[1,2]

[1]Institute of Applied Physics, University of Bern, Bern, Switzerland
[2]Oeschger Centre for Climate Change Research, University of Bern, Bern, Switzerland
[3]Karlsruhe Institute of Technology, Institute of Meteorology and Climate Research, Karlsruhe, Germany
[4]Centre National de la Recherche Scientifique, Université de Versailles Saint-Quentin-en-Yvelines, Guyancourt, France
[5]MeteoSwiss, Payerne, Switzerland
[6]Deutscher Wetterdienst, Hohenpeissenberg, Germany

**Correspondence:** Leonie Bernet (leonie.bernet@iap.unibe.ch)

**Abstract.** Observing stratospheric ozone is essential to assess if the Montreal Protocol has succeeded to save the ozone layer by banning ozone depleting substances. Recent studies have reported positive trends indicating that ozone is recovering in the upper stratosphere at mid-latitudes, but the trend magnitudes differ and uncertainties are still high. Trends and their uncertainties are influenced by factors such as instrumental drifts, sampling patterns, discontinuities, biases, or short-term anomalies that all might mask a potential ozone recovery. The present study investigates how anomalies, temporal measurement sampling rates and trend period lengths influence resulting trends. We present an approach for handling suspicious anomalies in trend estimations to improve the derived trend profiles. The approach was applied to data from a Ground-based Millimetre-wave Ozone Spectrometer (GROMOS) located in Bern, Switzerland. We compare our improved GROMOS trend estimate with results from other ground stations (lidars, ozonesondes, and microwave radiometers) in Central Europe. The data indicate positive trends of 1 to 3 % per decade at an altitude of about $40\,\mathrm{km}$ ($3\,\mathrm{hPa}$), providing a confirmation of ozone recovery in the upper stratosphere in agreement to satellite observations. At lower altitudes, the ground station data show inconsistent trend results, which emphasize the importance of ongoing research on lower stratospheric ozone trends. Our presented method of a combined analysis of ground station data provides a useful approach to recognize and to reduce uncertainties in stratospheric ozone trends by considering anomalies in the trend estimation. We conclude that stratospheric trend estimations still need improvement and that our approach provides a tool that can also be useful for other data sets.

## 1 Introduction

After the serious stratospheric ozone decrease due to ozone depleting substances (ODS) (Molina and Rowland, 1974; Chubachi, 1984; Farman et al., 1985), signs of an ozone recovery have been reported in recent years (e.g. WMO, 2014). Implementing the Montreal Protocol (1987) has succeeded to reduce ODS emissions so that the total chlorine concentration has been decreasing since 1997 (Jones et al., 2011). As a consequence, Antarctic ozone started to increase again, as shown by recent studies (Solomon et al., 2016; Kuttippurath and Nair, 2017; Pazmiño et al., 2018; Strahan and Douglass, 2018). Outside of the polar



regions, however, increasing ozone is more difficult to detect and depends on altitude and latitude. It is still controversial if ozone is recovering in the lower stratosphere (Ball et al., 2018; Chipperfield et al., 2018; Stone et al., 2018; Wargan et al., 2018), whereas broad consensus exists that stratospheric ozone has stopped to decline in the upper stratosphere since the end of the 1990s (Newchurch et al., 2003; Reinsel et al., 2005; Steinbrecht et al., 2006; Stolarski and Frith, 2006; Zanis et al.,

2006; Steinbrecht et al., 2009a; Shepherd et al., 2014; WMO, 2014). Recently estimated trends for upper stratospheric ozone are positive, but they are still different in magnitude and significance, because detecting a small trend is a difficult task. A trend might be masked by natural variability and many factors influence stratospheric ozone such as greenhouse gas increase and variations in atmospheric dynamics, solar irradiance or volcanic aerosols (Pawson et al., 2014). Other important sources for trend uncertainties are instrumental drifts, abrupt changes, biases, or sampling issues e.g. due to different vertical or temporal

resolution or different sampling patterns (Pawson et al., 2014; Damadeo et al., 2018). Satellite drifts have been included in trend uncertainties in several studies (e.g. Stolarski and Frith, 2006). Possible statistical methods to consider abrupt changes in a time series are for example presented by Bates et al. (2012). Bai et al. (2017) found that biases in ozone data sets can lead to important differences in trend estimates, especially when they are occurring in the beginning or the end of the considered trend period. The influence of non-uniform sampling patterns on trends was illustrated by Millán et al. (2016). Also Damadeo et al.

(2018) showed that accounting for temporal and spatial sampling biases and diurnal variability changes satellite based trends.

To account for several of the mentioned factors that influence trend estimates, different approaches were published following the Scientific assessment of Ozone Depletion of the World Meteorological Organization (WMO) in 2014 (WMO, 2014), with the aim to reduce uncertainties in trend estimates. Drifts in single satellite data sets were, for example, considered in the studies by Eckert et al. (2014) or Bourassa et al. (2018), whereas Sofieva et al. (2017) used only stable satellite products with small

drifts. The study of Steinbrecht et al. (2017) summarizes recent trend estimates using only updated satellite data sets that are drift corrected. The drifts were mainly identified by Hubert et al. (2016) and not yet considered in the trend estimates by Harris et al. (2015) or WMO (2014). Steinbrecht et al. also used data from a large range of ground stations, but possible biases or anomalies in these ground-based data were not considered. The resulting ground-based trends consequently show some important differences, and were not used in their final merged trend profile. Ball et al. (2017) used an advanced trend

estimation method that considers data steps in satellite time series or biases due to measurement artefacts. Their Bayesian method uses a priori information about the different satellite data sets and results in an optimal merged ozone composite, but it has not been applied yet to ground-based data sets.

All of these studies using corrected satellite data agree on positive ozone trends in the upper stratosphere with some differences in magnitude, and show varying trends in the middle and lower stratosphere. This agreement is more difficult to observe

in ground-based data sets, where the data variability is larger due to strong regional variability (Steinbrecht et al., 2017; Pawson et al., 2014). Because of this larger variability, considering instrumental biases or regional anomalies is of special importance for trend estimations derived from ground-based data. In addition to Steinbrecht et al. (2017) and WMO (2014), several other studies presented ground-based trends of stratospheric ozone profiles (e.g. Steinbrecht et al., 2009a; Nair et al., 2013, 2015; Harris et al., 2015), but biases in the data sets that might influence the resulting trends have not been considered yet.





The present study proposes an approach to handle the problem of anomalies in trend data sets by considering anomalies when estimating trends from ground-based data sets. For this purpose, we present the updated data set of the ground-based microwave radiometer GROMOS (**Gro**und-based **M**illimetre-wave **O**zone **S**pectrometer) located in Bern, Switzerland. We determine its trends with a multilinear parametric trend model (von Clarmann et al., 2010) by considering anomalies and

uncertainties in the time series, resulting in a corrected trend estimate. To identify biased periods, we compare the GROMOS data with other ground-based data sets (lidars, ozonesondes, and microwave radiometers) in Central Europe (Sect. 3). Before applying our trend approach to the GROMOS time series (Sect. 4.2), we tested it with an artificial time series (Sect. 4.1). Not only anomalies in a time series influence resulting trends, but also sampling patterns and the choice of the trend period. We therefore present a short analysis of temporal sampling rate and trend period length based on the GROMOS data set (Sect. 4.3

and 4.4). Finally, we compared the corrected GROMOS trend with the trends from the other data sets used (Sect. 4.5).

## 2    Data and methods

Our study presents how to consider anomalies in a time series to improve resulting trend estimates. We apply this approach to data from the GROMOS microwave radiometer and compare the time series and the trend with different ground-based ozone data sets.

### 2.1    Ozone data sets

The stratospheric ozone profile data used in the present study comes from different instruments that measure in Central Europe (France, Germany, and Switzerland). They are all part of the Network for the Detection of Atmospheric Composition Change (NDACC, 2018). In addition, we used data from the Microwave Limb Sounder (MLS) on board the Aura satellite (Aura/MLS). We consider data from Jan. 1995 to Dec. 2017, except for some instruments that cover a shorter time period (Table 1).

### 2.1.1    Microwave radiometers

We use data from two microwave radiometers, both located in Switzerland. They measure atmospheric emission at 142 GHz, where ozone molecules emit microwave radiation due to rotational transitions. The spectral line measured is pressure broadened and thus contains altitude information of the ozone molecules. To obtain a vertical ozone profile, the received radiative intensity is compared to the spectrum simulated by the Atmospheric Radiative Transfer Simulator 2 (ARTS2, Eriksson et al. (2011)).

By using an optimal estimation method according to Rodgers (2000), the best estimate of the vertical profile of ozone volume mixing ratio is then retrieved from the measured spectrum. This is done by the software tool Qpack2, which provides together with ARTS2 an entire retrieval environment (Eriksson et al., 2005).

The **Gro**und-based **M**illimetre-wave **O**zone **S**pectrometer (GROMOS) located in Bern, Switzerland (46.95° N, 7.44° E) at an altitude of 574 m above sea level (a.s.l.) is main focus of this study (Kämpfer, 1995; Peter, 1997). GROMOS has been

measuring continuously ozone spectra since November 1994. Before October 2009, the measurements were performed by means of a filter bench (FB) with an integration time of 1 h. Since October 2009, a Fast Fourier Transform Spectrometer





(FFTS) with an integration time of 30 min has been used. An overlap measurement period of almost 2 years (Oct. 2009 to July 2011) was used to homogenize the FB data, by subtracting the mean bias profile of the overlap period ($FB_{mean} - FFTS_{mean}$) from all FB profiles (Moreira et al., 2015). These homogenized ozone data are available on the NDACC web page (ftp.cpc.ncep. noaa.gov/ndacc/station/bern/hdf/mwave/). The FFTS retrieval used in the present study (version 2021) uses variable errors in

the a priori covariance matrix of around 30 % in the stratosphere and 70 % in the mesosphere, and a constant measurement error of 0.8 K (Moreira, 2017). The retrieved profiles have a vertical resolution of $\sim$ 15 to 25 km in the stratosphere. We concentrate in this study on the middle and upper stratosphere between 31 hPa ($\approx$ 24 km) and 0.8 hPa ($\approx$ 49 km), where the retrieved ozone is quasi independent of the a priori information. This is assured by limiting the altitude range to the altitudes where the area of the averaging kernels (measurement response) is larger than 0.8, which means that more than 80 % of the information

comes from the observation instead the a priori data (Rodgers, 2000). More information about the homogenization as well as the parameters used in the retrieval can be found in Moreira et al. (2015). Besides the described data harmonization to account for the instrument upgrade, we performed some additional data correction. Because the stratospheric signal is weak in an opaque and humid atmosphere, we discarded measurements when the atmospheric transmittance was smaller than 0.3. Excluding measurements in such a way should not result in a sampling bias because tropospheric humidity is uncorrelated

to stratospheric ozone. Also, the data have been corrected for outliers at each pressure level by removing values that exceed four times the standard deviation within a 3-days moving window. Profiles were excluded completely when more than 50 % of their values were missing (e.g. due to outlier detection). Furthermore, we omitted profiles where the instrumental system temperature showed outliers, identified by an exceed of four times the standard deviation within a 30-days moving window.

The second microwave radiometer used in this study is the **S**tratospheric **O**zone **Mo**nitoring **Ra**diometer (SOMORA). It

was built in 2000 as an update of the GROMOS radiometer and is located at the aerological station (MeteoSwiss) in Payerne (46.8° N, 7.0° E) at 491 m a.s.l. since 2002. Some instrumental changes were performed in the beginning of 2005 (front-end change) and in October 2010, when the acousto-optical spectrometer of SOMORA has been upgraded to an FFTS (Maillard Barras et al., 2015). The data have been harmonized to account for the spectrometer change. The instrument covers an altitude range from 25 to 60 km with a temporal resolution of 30 min to 1 h. In this study, we consider SOMORA data at an altitude

range between 18 hPa ($\approx$ 27 km) and 0.8 hPa ($\approx$ 49 km). For more information about SOMORA, refer to Calisesi (2003) concerning the instrumental setup and Maillard Barras et al. (2009, 2015) concerning the operational version of the ozone retrievals used in the present study.

### 2.1.2 Lidars

We use data from two **di**fferential **a**bsorption **l**idar (DIAL, Schotland (1974)) instruments in Germany and France. The in-

struments emit laser pulses at two different wavelengths, where one is absorbed by ozone molecules and the other is not. Comparing the backscattered signal at these two wavelengths provides information on the vertical ozone distribution in the atmosphere.

The lidar at the Meteorological Observatory Hohenpeissenberg (MOH, 47.8° N, 11.0° E, 980 m a.s.l.) in Germany has been operating since 1987, emitting laser pulses at 308 and 353 nm (Werner et al., 1983; Steinbrecht et al., 2009b). The lidar can only



retrieve ozone profiles during clear-sky nights due to scattering on cloud particles and the influence of sunlight. On average, it retrieves eight night-profiles in a month. In this study we limit the data to the altitude range where the averaged measurement error is below 10 % (below 43 km or 2 hPa). The lower altitude limit was set to the chosen limit of GROMOS at 31 hPa ($\approx$ 24 km).

The Observatory of Haute Provence (OHP, 43.9° N, 5.7° E, 650 m a.s.l.) in France hosts a lidar that measures in its current setup since the end of 1993 (Godin-Beekmann et al., 2003). The lidar emits laser pulses at 308 nm and 355 nm, as first described by Godin et al. (1989). The instrument measures on average 11 profiles per month. We use OHP lidar profiles below 40 km ($\sim$ 3 hPa), where the averaged measurement error remains below 10 %. As lower altitude limit we use 31 hPa ($\approx$ 24 km) to be consistent with the GROMOS limits.

More detailed information about the lidars and ozonesonde used can be found for example in Godin et al. (1999) and Nair et al. (2011, 2012).

### 2.1.3    Ozonesondes

The three mentioned observatories at Payerne, MOH and OHP also provide weekly ozonesonde measurements. The ozonesonde measurements in Payerne are usually performed three times a week at 11:00 UTC (Jeannet et al., 2007), resulting in 13 pro-
files per month on average. The meteorological balloon carried a Brewer Mast sonde (BM, Brewer and Milford (1960)) until September 2002, which was then replaced by an electrochemical concentration cell (ECC, Komhyr (1969)). The profiles are normalised using total column ozone from the Dobson spectrometer at Arosa (46.77° N, 9.7° E, 1850 m a.s.l., Favaro et al. (2002)). If those data are not available, forecast ozone column estimates based on GOME–2 (Global Ozone Monitoring Experiment–2) data are used (http://www.temis.nl/uvradiation/nrt/uvindex.php).

Ozone soundings at MOH are performed two to three times per week with a BM sonde (on average 10 profiles per month). Three different radiosonde types have been used since 1995, all carrying a BM ozonesonde, without major changes in its performance since 1974 (Steinbrecht et al., 2016). The profiles are normalized by on-site Dobson or Brewer spectrophotometers and, if not available, by satellite data (Steinbrecht et al., 2016).

At OHP, ECC ozonesondes have been used since 1991 with several instrumental changes (Gaudel et al., 2015). The data
are normalized with total column ozone measured by a Dobson spectrophotometer until 2007 and ultraviolet-visible SAOZ (Système d'Analyse par Observation Zénithale) spectrometer afterwards (Guirlet et al., 2000; Nair et al., 2011). In our analysed period, four profiles are on average available per month.

Ozonesonde data are limited to altitudes up to $\sim$ 30 km, where the balloon usually bursts. Therefore, we used ozonesonde profiles only below 30 km, which is a threshold value for Brewer Mast ozonesondes with precision and accuracy below $\pm$5 %
(Smit and Kley, 1996). For normalization, the correction factor (CF), which is the ratio of total column ozone from the reference instrument to the total ozone from the sonde, has been applied to all ozonesonde profiles. At all measurement stations, we discarded profiles when their CF was larger than 1.2 or smaller than 0.8 (Harris et al., 1998; Smit and ASOPOS Panel, 2013). We further excluded negative values and profiles with extreme jumps or constant ozone values, as well as profiles with constant or decreasing altitude values.



### 2.1.4 Aura/MLS

The microwave limb sounder (MLS) on the Aura satellite, which was launched in mid 2004, measures microwave emission from the Earth in five broad spectral bands (Parkinson et al., 2006). It provides profiles of different molecules in the atmosphere, including stratospheric ozone, with a vertical resolution of $\sim 3\,\mathrm{km}$. We used ozone data from Aura/MLS version 4.2 above

Bern with a spatial coincidence of $\pm 1\,^\circ$ latitude and $\pm 8\,^\circ$ longitude, where the satellite passes twice a day (around 02:00 and 13:00 UTC). More information about the MLS instrument and the data product can be found, e.g., in Waters et al. (2006). We chose the Aura/MLS data for our study because it was identified to be stable with insignificant drifts between 20 and $40\,\mathrm{km}$ (Hubert et al., 2016).

### 2.2 Data comparison

To verify the GROMOS data and detect possible anomalies, we compared all described data sets with the GROMOS data from Bern in the time period Jan. 1995 to Dec. 2017. The different instrument sites are all located in Central Europe. Payerne is located $40\,\mathrm{km}$ south-west of Bern, which ensures comparable stratospheric measurements. Hohenpeissenberg is located $290\,\mathrm{km}$ north-east of Bern, and the OHP lies $360\,\mathrm{km}$ south-west of Bern. Even if stratospheric trace gases generally show small horizontal variability, the distance between the different stations, especially between MOH and OHP, may lead to some

differences in measured ozone.

### 2.2.1 Data processing for comparison

The GROMOS retrieval is accurate between 31 and $0.8\,\mathrm{hPa}$. We therefore limit all instrument data in this study to this altitude range, and divide it in three parts. For convenience, they will be referred to as lower stratosphere between 31 and $13\,\mathrm{hPa}$ ($\approx 24$ to $29\,\mathrm{km}$), middle stratosphere between 13 and $3\,\mathrm{hPa}$ ($\approx 29$ to $39\,\mathrm{km}$) and upper stratosphere between 3 and $0.8\,\mathrm{hPa}$ ($\approx 39$ to

$49\,\mathrm{km}$). The limits for the upper stratosphere agree with the common definition (e.g. Ramaswamy et al., 2001), whereas the here defined lower and middle stratosphere are usually referred to as middle stratosphere in other studies.

Most of the instruments provide volume mixing ratios (VMR) of ozone in parts per million (ppm). In case that the data were given in number density ($\mathrm{molecules\,cm^{-3}}$), as for example for the OHP ozonesonde, the VMR was calculated with the air pressure and temperature provided by the same instrument or collocated radiosonde data. Data that can not be physically

explained such as negative ozone values have been excluded for all instruments.

The GROMOS, SOMORA and Aura/MLS profiles have a constant pressure grid, which is not the case for the lidar and ozonesonde data. We therefore interpolated all lidar and ozonesonde profiles to a pressure grid based on a regular spaced altitude grid of $100\,\mathrm{m}$ for the ozonesonde at OHP and Payerne, and $300\,\mathrm{m}$ for the lidars and the ozonesonde at MOH. The mean profile of the interpolated pressure data then built the new pressure grid for the ozone data. These interpolated lidar and

ozonesonde data are used for the trend estimations. For the direct comparison with GROMOS, the data were adapted to the GROMOS grid, which is described in the next section (Sect. 2.2.2). Our figures generally show both pressure and geometric



altitude. The geometric altitude is approximated by the mean altitude grid from GROMOS, which is for each retrieved profile determined from operational model data of the European Centre for Medium-Range Weather Forecasts (ECMWF).

### 2.2.2 Direct comparison with GROMOS

The vertical resolution of GROMOS is usually coarser than for the other instruments, except for SOMORA. When compar-
ing their profiles directly with GROMOS profiles, the different vertical resolution of the instruments has to be considered. Smoothing the profiles of the different instruments by GROMOS' averaging kernels makes it possible to compare the profiles with GROMOS without biases due to resolution or a priori information (Tsou et al., 1995). The profiles having higher vertical resolution than GROMOS were interpolated to the GROMOS pressure grid, and were then convolved by the averaging kernel matrix according to Connor et al. (1991), with

$$x_{conv} = x_a + \mathbf{AVK}(x_h - x_a) \tag{1}$$

where $x_{conv}$ is the resulting convolved profile, $x_a$ is the a priori profile used in the GROMOS retrieval, $x_h$ is the profile of the higher resolved instrument, interpolated to the GROMOS grid, and $\mathbf{AVK}$ is the corresponding averaging kernel matrix from GROMOS. When $x_h$ covered a smaller altitude range than GROMOS, the profile $x_h$ was extended by the a priori profile of GROMOS to cover the same altitude range. After the convolution, these additional values were removed again to obtain the
original altitude range of $x_h$. The SOMORA profiles have a similar vertical resolution as profiles from GROMOS and were thus not convolved, because this would require a more advanced comparison method as proposed by Rodgers and Connor (2003) or Calisesi et al. (2005). GROMOS has a higher temporal resolution than the other instruments (except SOMORA). For SOMORA, only profiles coincident in time with GROMOS have been selected. For the other instruments, a mean of GROMOS data at the time of the corresponding measurement was used, with a time coincidence of $\pm 30$min. Only for the
lidars, GROMOS data were averaged over the whole lidar measurement time (usually one night).

For direct comparison with GROMOS we computed relative differences between the monthly mean values of the different data sets and the mean of the coincident GROMOS profiles. The relative differences have been calculated by subtracting each monthly ozone value of the data set X from the corresponding GROMOS monthly mean (GR), using the GROMOS monthly mean of the corresponding month as a reference $((GR - X)/GR \cdot 100)$. Based on these relative differences we identified periods
where GROMOS showed biased values compared to the other instruments. To identify these anomalies we used a corrected relative difference to GROMOS. For this the relative difference time series for each instrument was corrected by its mean relative difference to GROMOS. This made it possible to ignore a potential constant offset of the instruments and to concentrate on periods with temporally large differences to GROMOS. When this corrected relative difference was larger than $10\,\%$ for at least three instruments, the respective month was identified as anomaly. Above $2\,$hPa, where only SOMORA and Aura/MLS
data are available, both data sets need to have a corrected relative difference larger than $10\,\%$ to be identified as anomaly.





## 2.3 Trend model

To estimate stratospheric ozone trends we applied the multilinear parametric trend model from von Clarmann et al. (2010) to the monthly means of all individual data sets. The model fits the following regression function:

$$
\begin{aligned}
y(t) = a &+ b \cdot t \\
&+ c \cdot \text{QBO}_{30\,\text{hPa}}(t) + d \cdot \text{QBO}_{50\,\text{hPa}}(t) \\
&+ e \cdot \text{F10.7}(t) \\
&+ f \cdot \text{MEI}(t) \\
&+ \sum_{n=1}^{4} \left( g_n \cdot \sin\left(\frac{2\pi t}{l_n}\right) + h_n \cdot \cos\left(\frac{2\pi t}{l_n}\right) \right)
\end{aligned}
\tag{2}
$$

where $t$ is the monthly time vector, $y(t)$ represents the estimated ozone time series, and $a$ to $h$ are coefficients that are fitted in the trend model. $\text{QBO}_{30\,\text{hPa}}$ and $\text{QBO}_{50\,\text{hPa}}$ are the normalized Singapore winds at 30 and 50 hPa, used as indices of the quasi–biennial oscillation (QBO, available at http://www.geo.fu-berlin.de/met/ag/strat/produkte/qbo/singapore.dat). F10.7 is the solar flux at a wavelength of 10.7 cm used to represent the solar activity (measured in Ottawa and Penticton, Canada (National Research Council of Canada, 2018)). The El Niño Southern Oscillation (ENSO) is considered by the Multivariate ENSO Index (MEI), that combines six meteorological variables measured over the tropical Pacific (Wolter and Timlin, 1998). The F10.7 data and the MEI data are available via https://www.esrl.noaa.gov/psd/data/climateindices/list/. In addition to the described natural oscillations, we used four periodic oscillations with different wavelengths $l_n$ to account for annual ($l_1 = 12$ months) and semi-annual ($l_2 = 6$ months) oscillation as well as two further overtones of the annual cycle ($l_3 = 4$ months and $l_4 = 3$ months).

The strength of the model used is that it can consider inhomogeneities in data sets, by considering a full error covariance matrix ($\mathbf{S_y}$) when reducing the cost function ($\chi^2$). Inhomogeneous data can originate from changes in the measurement system (e.g. changes in calibration standards or merging of data sets with different instrumental modes), or from irregularities in spatial or temporal sampling. Such inhomogeneities lead to groups of temporal correlated data, that can, if not considered, change significance and slope of the estimated trend (von Clarmann et al., 2010). Von Clarmann et al. present an approach to consider such inhomogeneities in the trend analysis. They divide the data into multiple subsets which are assumed to be biased with respect to each other and which are characterized by diagonal blocks in the data covariance matrix. Another reason for inhomogeneities might be unknown instrumental issues that lead to anomalies, which is treated in the present study.

To consider inhomogeneities, the total uncertainty of the data set is represented by a full error covariance matrix $\mathbf{S_y}$ at each pressure level. The diagonal elements of $\mathbf{S_y}$ are set to the monthly means of the measurement uncertainties for each instrument. For lidar data this is on average 4 % for the OHP lidar and 6 % at MOH between $\sim 20$ and $40\,\text{km}$, with lowest uncertainties in the middle stratosphere. For the ozonesonde, the uncertainty is assumed to be 5 % for all ECC sondes and 10 % for BM sondes (Harris et al., 1998; Smit and ASOPOS Panel, 2013). For SOMORA we use uncertainties composed of smoothing and observational error, ranging from 1 to 2 % in the middle stratosphere and 2 to 7 % in the upper stratosphere. The used Aura/MLS uncertainties range between 2 and 5 % throughout the stratosphere. For the GROMOS trends, we use



uncertainty estimates as described by Moreira et al. (2015), composed of the standard error ($\sigma/\sqrt{\mathrm{DOF}}$ with standard deviation $\sigma$ and degrees of freedom DOF) of the monthly means, an instrumental uncertainty (measurement noise), and an estimated systematic instrument uncertainty obtained from cross-comparison. The resulting uncertainty values are approximately 5 % in the middle stratosphere, 6 to 8 % in the lower stratosphere and 6 to 7 % in the upper stratosphere. The off-diagonal elements of

$\mathbf{S_y}$ are first set to zero, assuming no error correlation. In a second step, autocorrelation coefficients are inferred from residuals of the initial trend fit. These are used to build another covariance matrix which is added to the data error covariance matrix. The mean variance of this additional covariance matrix is adjusted such that the $\chi^2$ divided by the degrees of freedom of the trend fit with the enhanced covariance matrix is unity. This additional covariance matrix represents contributions to the fit residuals which are not caused by data errors but by phenomena that are not represented by the trend model. This second

trend fit is only performed if the initial normalized $\chi^2$ is larger than unity, which is not the case if the assumed data errors are larger than the fit residuals (Stiller et al., 2012). The more sophisticated uncertainty estimates that we use for GROMOS are larger than the residuals in the first regression fit ($\chi^2 < 1$), which means that the correlated residuals are not considered for the GROMOS trend. For all the other instruments, however, correlated residuals are considered, because we use simple instrumental uncertainties that are usually smaller than the fitted residuals.

To account for anomalies in the time series when estimating the trends we adapted $\mathbf{S_y}$ in two steps. First, we increased the uncertainties for months where anomalies were identified. For this purpose, we set the diagonal elements of $\mathbf{S_y}$ for the respective months to a value obtained from the mean difference to all instruments and the mean GROMOS ozone value at each altitude. Assuming for example that GROMOS deviates from all instruments on average by 10 % in August 2010 at 10 hPa, and the overall mean August value at this altitude would be 7 ppm. In this case we would assume an uncertainty of 0.7 ppm for this

biased month at this altitude level. In a second step, we used the ability of the trend programme to account for biases in a data subset. For this, a fully correlated block composed of the squared estimated bias uncertainties is added to the corresponding part of the covariance matrix. Considering the bias in this way is mathematically equivalent to treating the bias as an additional fit variable that is fitted to the regression model with an optimal estimation method, as shown by von Clarmann et al. (2001). The value chosen for the bias uncertainty determines how much the bias is estimated from the data itself. For low values, the

bias will be close to the a priori value, which would be a bias of zero, for high values it will be estimated completely from the given data. The bias can thus be estimated from the data itself, which makes the method more robust because it does not depend on an a priori choice of the bias (Stiller et al., 2012). In a sensitivity test we have found that assuming a value of 5 ppm for the correlated block permits a reliable fit, where the bias is fitted independently of the a priori zero bias.

Our ozone trend estimates always start in Jan. 1997, which is the most likely turning point for ozone recovery due to the

decrease of ODS (Jones et al., 2011), and all end in Dec. 2017. Exceptions are the trends from SOMORA and Aura/MLS. The SOMORA trend starts in Jan. 2000 when the instrument started to measure ozone and the Aura/MLS trend starts in Jan. 2005. Aura/MLS trends build the shortest trend period, starting only in Jan. 2005. The trends are always given in % per decade, which is determined from the model output in ppm per decade by dividing it by the overall ozone mean value of the corresponding data set at each altitude level. We declare a trend to be significantly different from zero at 95 % confidence interval, as soon as





its absolute value exceeds twice its uncertainty, as described by Tiao et al. (1990). This statistically inferred confidence interval is based on the assumed instrumental uncertainties. Unknown drifts of the data sets, however, are not considered in this claim.

## 3 Time series comparison

To verify the GROMOS data and identify potential anomalies, we first have a look at the GROMOS time series (Sect. 3.1) and
compare then the data with instrument data in Central Europe (Sect. 3.2).

### 3.1 GROMOS time series

The monthly means of the GROMOS time series (Fig. 1 (a)) clearly depict the maximum of ozone VMR between $10\,\mathrm{hPa}$ and $5\,\mathrm{hPa}$ and the seasonal ozone variation, with increased spring–summer ozone in the middle stratosphere and increased autumn–winter values in the upper stratosphere (Moreira et al., 2016). Figure 1 (b) shows GROMOS' relative deviations from
the monthly climatology (monthly means over the whole period 1995 to 2017). This ozone anomaly is calculated by the ratio of the deseasonalised monthly means (difference between each individual monthly mean and the corresponding climatology of this month) and the monthly climatology. We observe some periods where GROMOS data deviates from its usual values, mostly distinguishable between the lower/middle stratosphere and the upper stratosphere. In the lower/middle stratosphere we observe negative anomalies (less ozone than usual) in 1995 to 1997 and 2005 to 2006 and positive anomalies (more ozone
than usual) in 1998 to 2000, in 2014 to 2015 and in 2017. In the upper stratosphere the data show negative anomalies in 2016 and positive anomalies in 2000 and 2002 to 2003. Strong but short-term positive anomalies are visible in 1997 in the upper stratosphere and in the beginning of 2014. The positive anomaly in the upper stratosphere in 1997 is due to some missing data in November 1997 because of an instrumental upgrade, leading to a higher monthly mean value than usual. Besides this we did not detect any systematic instrumental issues in the GROMOS data that could explain the anomalies. Therefore, we compare
the GROMOS data with the presented ground-based instruments from Central Europe, as well as with Aura/MLS data, to check whether the observed anomalies are due to natural variability or due to unexplained instrumental issues.

### 3.2 Comparison of different data sets

We compared GROMOS with ground-based and Aura/MLS data and averaged it over three altitude ranges (Fig. 2). The different data sets have been smoothed with the averaging kernels of GROMOS to make a direct comparison possible, as
described in Sect. 2.2.2. Due to the similar vertical resolution of GROMOS and SOMORA, the SOMORA profiles have not been smoothed by GROMOS' averaging kernels, despite differences between their a priori data and averaging kernels. This might lead to larger differences between GROMOS and SOMORA than between GROMOS and the other instruments. To avoid that an instrument does not cover the full range of one of the three altitude classes, all ozonesonde data have been cut at $30\,\mathrm{km}$ ($\approx 11.5\,\mathrm{hPa}$), all lidar data at $3\,\mathrm{hPa}$ ($\approx 40\,\mathrm{km}$), and all SOMORA data below $13\,\mathrm{hPa}$ ($\approx 29\,\mathrm{km}$) for this analysis.
The different instrument time series shown in Fig. 2 contain only data that are coincident with GROMOS measurements as





described in Sect. 2.2.2, whereas the here shown GROMOS data represent the complete time series with another, usually finer temporal sampling. This might lead to some sampling differences that are not considered in this figure.

The different data sets agree well, showing, however, periods where some instruments deviate more from GROMOS than others. In the upper stratosphere (Fig. 2 (a)), GROMOS and SOMORA agree well most of the time, but GROMOS reports

slightly lower ozone than SOMORA and also lower values than Aura/MLS. In the middle stratosphere (Fig. 2 (b)), both lidars exceed the other instrument data in the last years, starting in 2004 at OHP and in 2010 at MOH. Similar deviations of the MOH and OHP lidars have also been observed by Steinbrecht et al. (2017), as can be seen in the latitudinal lidar averages of their study.

Differences between the data sets in the lower stratosphere can be better seen in Fig. 3. Shown are the monthly relative

differences of time coincident pairs of GROMOS (GR) and the convolved data set X, with GROMOS data as reference value ($(GR - X)/GR \cdot 100$). The mean relative difference of all instruments compared to GROMOS (black line in Fig. 3) generally lies within $\sim \pm 10\,\%$. However, there are some periods with larger deviations, where GROMOS measures lower ozone than the other instruments (negative relative difference) in 1995 to 1997 and in 2006 in the lower stratosphere, and in 2016 in the middle and upper stratosphere. We further observe that the relative difference between GROMOS and the OHP ozonesonde

shows some important peaks in the last decade, indicating that the sonde often measures higher ozone than GROMOS. This can be explained by the low number of OHP ozonesonde profiles per month (only four profiles on average), which leads occasionally to large differences in the monthly mean when comparing only few profiles with time coincident GROMOS data.

For a broader picture, the same relative differences to GROMOS are shown in Fig. 4, but each panel represents an individual instrument and all altitude levels are shown. The anomalies where at least three data sets (or two above $2\,\mathrm{hPa}$) deviate by more

than $10\,\%$ from GROMOS (as described in Sect. 2.2.2) are shown in the lowest panel in black. In addition to the negative anomalies observed already in the other figures (e.g. in 2006 in the lower stratosphere and in 2016 in the upper stratosphere), we also observe positive anomalies in the lower/middle stratosphere in 2000, 2014, 2015, and 2017. The negative anomalies in 1995 to 1997 in the lower stratosphere that we observed in Fig. 3 were only partly detected as anomalies with our anomaly criteria.

To summarize our comparison results, we observed some periods with anomalies compared to GROMOS' climatology (Fig. 1). Some of these anomalies were also observed when comparing GROMOS to the different data sets (Fig. 2, Fig. 3, and Fig. 4). This implies that the source of the anomalies are local variations in Bern or instrumental issues of GROMOS rather than broad atmospheric variability. We can thus conclude that the observed negative GROMOS anomalies in the lower stratosphere in 2006 and in the upper stratosphere in 2016 are biases in the GROMOS time series. The same is the case for the positive

anomalies in 2000, 2014, 2015 and 2017 in the lower and middle stratosphere. In contrast to these confirmed anomalies, the GROMOS anomalies in the lower stratosphere in 1995 to 1997 (negative) and 1998 and 1999 (positive) as observed in Fig. 1 are small when comparing to the other instruments and are thus only confirmed for a few months by our anomaly detection. The biased periods in the upper stratosphere in 2000 and 2002 to 2003 (positive anomalies) were not confirmed by comparing GROMOS to the other data sets, and might thus be real ozone variations. However, we have to keep in mind that less

instruments (only SOMORA and Aura/MLS) provide data for comparison above $2\,\mathrm{hPa}$, which makes the anomaly detection





less robust at these altitudes. Our results are consistent with those reported by Moreira et al. (2017). They compared GROMOS with Aura/MLS data and also observed positive deviation of GROMOS in the middle stratosphere in 2014 and 2015, as well as a negative deviation in the upper stratosphere in 2016. Hubert et al. (2018) found similar anomalies in the GROMOS time series by comparing ground-based data sets to several satellite products (see also Petropavlovskikh et al. (2018)). Some of our

detected biased periods were also found by Steinbrecht et al. (2009a) who compared different ground-based instruments, for example the GROMOS anomaly in 2006. They attribute the observed biases to sampling differences, but also to irregularities in some data sets. In fact, our results confirm irregularities in the GROMOS time series, which are probably due to instrumental issues of GROMOS.

## 4 Ozone trend estimations

By comparing the GROMOS data to the other instruments, we have confirmed some biased periods in the GROMOS time series. To improve the GROMOS trend estimates, we use now these detected anomalies by considering them in the regression fit (Sect. 4.2). In the following section, this method will first be applied to an artificial time series to test and illustrate the approach.

### 4.1 Artificial time series analysis

The trend programme from von Clarmann et al. (2010) can handle uncertainties in a flexible way, which makes it possible to account for anomalies in a time series. To investigate how anomalies can be best considered in the trend programme, we test the programme with a simple artificial time series that consists of monthly ozone values with a seasonal cycle and a linear trend. This artificial time series $y(t)$ is given by

$$y(t) = a + b \cdot t + g \cdot \sin\left(\frac{2\pi t}{l_n}\right) + h \cdot \cos\left(\frac{2\pi t}{l_n}\right) \tag{3}$$

with the monthly time vector $t$, a constant ozone value $a = 7\,\mathrm{ppm}$ and a trend of $0.1\,\mathrm{ppm}$ per decade, with $b = 0.1/120\,\mathrm{ppm}$ per month. We consider a simple seasonal oscillation with an amplitude $A = -1\,\mathrm{ppm}$ such that $A^2 = g^2 + h^2$ (e.g. $g = h = \sqrt{0.5}\,\mathrm{ppm}$) and a wavelength $l_n = 12$ months. The uncertainty was assumed to be $0.1\,\mathrm{ppm}$ for each monthly ozone value, which was considered in the diagonal elements of $\mathbf{S_y}$. This artificial time series (shown in Fig. 5 (a)) is later on referred to as case A. The estimated trend for this simple time series corresponds quasi perfectly to the assumed time series' trend ($0.1\,\mathrm{ppm}$

per decade), which proves that the trend programme works well. The residuals (Fig. 5 (b)) are small ($-6$ order of magnitudes), but increase towards the time series edges. This might be due to some error propagation, but can be neglected for this study because of its small magnitude.

To investigate how the trend programme reacts to anomalies in the time series, we increased the monthly ozone values by 5 % ($\approx 0.4\,\mathrm{ppm}$) in the summer months (June, July, Aug.) of 2014, 2015 and 2017. When estimating the trend for this modified

time series (case B), we observe a trend of $\sim 0.13\,\mathrm{ppm}$ per decade instead of the expected $\sim 0.1\,\mathrm{ppm}$ per decade (Fig. 6). Assuming that a real time series contains such suspicious anomalies due to for example instrumental issues, they would distort

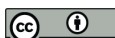



the true trend. To account for these anomalies in the trend estimation, we applied different possible corrections by modifying the error covariance matrix $\mathbf{S_y}$ in the trend model. The used parameters for the different corrections are summarized in Table 2, and Fig. 6 presents the resulting linear trends. In a first attempt we decreased the weight of the anomalies in the time series by increasing their uncertainties (diagonal elements of $\mathbf{S_y}$) and set the uncertainties of the corresponding months to 5 % of the

overall mean ozone value (case C). Afterwards, we added a correlated block to the covariance matrix for the months containing anomalies, and applied the $\mathbf{S_y}$ once with (case D) and once without (case E) the increased diagonal elements of $\mathbf{S_y}$. The bias uncertainty in the correlated block was set to 5 ppm.

     The results show that all of the corrected trend estimations (cases C, D, and E) improve the trends and lead to almost the true trend of 0.1 ppm (Fig. 6 (b)). They are thus able to correct for the artificially added anomalies. Considering a correlated

bias block in the $\mathbf{S_y}$ (case D and E) leads to even better results than only increasing the diagonal elements of $\mathbf{S_y}$ (case C). Consequently, the trends are improved by correcting the anomalies in the trend estimation, whereas the uncorrected estimate distorts the trend unintentionally. We further found that the trend estimate is closer to the true trend, the higher the uncertainties are chosen (diagonal elements of $\mathbf{S_y}$ or correlated block in $\mathbf{S_y}$). This can be explained by the fact that the trend analysis allows the bias to vary as in an optimal estimation scheme (von Clarmann et al., 2001). When estimating the bias, the higher the

bias uncertainty is, the less confidence is accounted to the a priori knowledge about the bias (that would be a bias of zero), and the bias is then determined directly from the data as if it would be an additionally fitted variable. Based on our test with the artificial time series we conclude that our method succeeds to handle suspicious anomalies in a time series, leading to an estimated trend close to the trend that would be expected without anomalies.

## 4.2    GROMOS trends

The GROMOS time series has been used for trend estimations in Moreira et al. (2015), who found a significant positive trend in the upper stratosphere. In recent years, GROMOS showed some anomalies leading to higher trends than expected in the middle atmosphere, as for example shown by Steinbrecht et al. (2017) or Petropavlovskikh et al. (2018). These higher trends motivated us to improve GROMOS trend estimates by accounting for the observed anomalies. In the following, we present the trend profiles of the GROMOS time series by considering the detected anomalies in the trend programme with the different

correction methods that were introduced in Sect. 4.1. In a first step, we estimated the trend without considering inhomogeneities in the data (case I in Fig. 7). The uncertainties of the data that are used as diagonal elements in $\mathbf{S_y}$ range between 5 and 8 % and are composed as proposed by Moreira et al. (2015) (see Sect. 2.3). In a second step, we increased the uncertainties (diagonal elements of $\mathbf{S_y}$) as described in Sect. 2.3 for the months and altitudes that were detected as anomalies, but still assume an uncorrelated error (case II). In a third step, we considered a full correlated block for the periods where anomalies were detected

to fit a bias (case III). The bias uncertainty was set to 5 ppm at all altitudes, which ensures that the bias is fully estimated from the data. The diagonal elements of $\mathbf{S_y}$ stayed the same as in case II.

     The trend profiles in Fig. 7 report an uncorrected GROMOS ozone trend (case I) of about 3 % per decade in the middle stratosphere from 30 to 5 hPa, which decreases above to around $-4$ % per decade at 0.8 hPa. Correcting the trend by increasing the uncertainty for months with anomalies (case II) decreases the trend slightly in the lower stratosphere, but the differences





are small. Using a correlated block in the covariance matrix to estimate the bias of each anomaly in an optimized way (case III) decreases the trend by around 1 % per decade in the lower stratosphere. The trend profile of this optimized trend estimation has a trend of 2 % per decade in the lower and middle stratosphere and peaks at approximately 4 hPa with a trend of 2.5 % per decade. The corrected trend profile agrees well with recent satellite-based ozone trends (e.g. Steinbrecht et al., 2017) between

∼ 20 and 40 km. In the upper stratosphere it decreases again, to −2.6 % per decade at 0.8 hPa.

All these different GROMOS trend estimates, considering anomalies in different ways, show a significant positive trend of ∼ 2.5 % per decade at around 4 hPa (≈ 37 km) altitude, and a trend decrease above. This agreement indicates that the trend at these altitudes is only marginally affected by the identified anomalies and rather robust. We can thus conclude that the trend of around 2.5 % per decade is a sign for an ozone recovery in the altitude range of 35 to 40 km. This result is consistent with

trends derived from merged satellite data sets as for example found in Ball et al. (2017), Frith et al. (2017), Sofieva et al. (2017) or Steinbrecht et al. (2017), even if the GROMOS trend peak is observed at slightly lower altitudes. A possible reason for this difference in the trend peak altitude might be related to the averaging kernels of the current GROMOS retrieval version. We observe a slight shift in the peak height of the averaging kernels, indicating that the information is retrieved from slightly higher altitudes (∼ 2 to 4 km). Furthermore, one has to keep in mind the coarse altitude resolution of GROMOS, which is at

this altitude between 20 to 25 km. It is therefore possible that the trend peak altitude does not exactly correspond to the true peak altitude. The instrumental upgrade in 2009 led to a change in the averaging kernels. This change, however, should not influence the trend estimates because the data have been harmonized (see Sect. 2.1.1) and thus corrected for such effects. The harmonization also corrects for possible effects due to changes in the temporal resolution (from 1 h to 30 min).

In the upper stratosphere (above 2 hPa), the GROMOS trend estimates are mostly insignificant but negative. This is probably

influenced by the negative trend observed in the mesosphere. A negative ozone trend in the mesosphere is consistent with theory, because increased methane emissions lead to an enhanced ozone loss cycle above 45 km (Pawson et al., 2014; Brasseur and Solomon, 2005). However, this is not further investigated in the present study because of the low mesospheric measurement response in the GROMOS filter bench data (before 2009).

In the middle and lower stratosphere (below 5 hPa) using different anomaly corrections results in largest trend differences,

with trends ranging from 2 to 3 % per decade. This result suggests that the GROMOS anomalies mostly affect these altitudes between 30 and 5 hPa. The corrected GROMOS trend (case III) is not significantly different from zero below 17 hPa, and cases I (uncorrected) and II (corrected by $\mathbf{S_y}$) show significant trends. Compared to other studies, the GROMOS trends in the lower stratosphere are slightly higher than trends of most merged satellite data sets, but the corrected GROMOS trend lies within their uncertainty bars.

In summary, correcting the GROMOS trend with our anomaly approach leads to a trend profile of 2 to 2.5 % per decade in the lower and middle stratosphere, which agrees with satellite-based trends from recent studies. The GROMOS trends are almost not affected by anomalies at 4 hPa (≈ 37 km), suggesting a robust ozone recovery of 2.5 % per decade. At lower altitudes, trends are more affected by the detected anomalies, and the corrected trend estimate shows a trend of 2 % per decade. This result implies that ozone might also recover at lower altitudes, but the uncertainties, the dependency on anomalies, and the

insignificance of the corrected trend show that this result is less robust.



### 4.3 Influence of temporal sampling on trends

Stratospheric ozone at northern mid-latitudes has a strong seasonal cycle of $\sim 16\,\%$ (Moreira et al., 2016) and a moderate diurnal cycle of 3 to 6 % (Schanz et al., 2014; Studer et al., 2014). The sampling rate of ozone data is thus important for trend estimates, because measurement dependencies on season or time of the day might influence the resulting trends. An important

characteristic of microwave radiometers is their measurement continuity, being able to measure during day and night, as well as during almost all weather conditions (except for an opaque atmosphere) and thus during all seasons. Other ground-based instruments such as lidars are temporally more restricted because they typically acquire data during clear-sky nights only. Clear-sky situations are more frequent during high-pressure events which vary with season and location (Steinbrecht et al., 2006; Hatzaki et al., 2014). The lidar measurements thus do not only depend on the daily cycle, but also on location and

season. The seasonal dependency for ozonesondes might be smaller, but they are only launched during daytime.

  Figure 8 gives an example of how the measurement sampling rate can influence resulting trends. The GROMOS time series is used for these trend estimates, once using only measurements at the time of ozonesonde launches in Payerne, and once only at the time of lidar measurements at MOH. The differences to the trend that uses the complete GROMOS sampling are not significant, but still important, especially at $\sim 40\,\mathrm{km}$ and in the lower stratosphere. The lidar sampling leads to higher trends

($\sim 3\,\%$ per decade) than the normal sampling ($\sim 2\,\%$ per decade) below $5\,\mathrm{hPa}$. Above $5\,\mathrm{hPa}$, using ozonesonde sampling results in a higher trend than using normal or lidar sampling. Even if ozonesondes are not measuring at those altitudes, the result shows that measuring with an ozonesonde sampling (e.g. only at noon) might influence the trend at these altitudes. Our findings suggest that the time of the measurement (day or night) or the number and the timing of measurements within a month can influence the resulting trend estimates. Results concerning the time dependences of trends based on SOMORA data can be

found in Maillard Barras et al. (2018). We conclude that sampling differences have to be kept in mind when comparing trend estimates from instruments with different sampling rates or measurement times.

### 4.4 Influence of time period on trends

An important factor that influences trend estimates is the length and starting year of the trend period. Several studies have shown that the choice of starting or end point affects substantially the resulting trend (Harris et al., 2015; Weber et al., 2018).

Further, the number of years in the trend period are crucial for the trend estimate (Vyushin et al., 2007; Millán et al., 2016). We investigate how the GROMOS trends change when the estimations start in different years. For this, we average the GROMOS trends over three altitude ranges and determine the trend for periods of different lengths, all ending in Dec. 2017 but starting in different years (Fig. 9). As expected, the uncertainties increase with decreasing period length, and trends starting after 2010 are thus not even shown. Consequently, trends become insignificant for short trend periods. In the lower and middle stratosphere,

more than 11 years are needed to detect a significant positive trend (at $95\,\%$ confidence level) in the GROMOS data, whereas the 23 years considered are not enough to detect a significant trend in the upper stratosphere (above $3\,\mathrm{hPa}$). Weatherhead et al. (2000) and Vyushin et al. (2007) state that at least 20 to 30 years are needed to detect a significant trend at mid-latitudes, but their results apply to total column ozone, which can not directly be compared with our ozone profiles. In general we observe




that the magnitude of the trend estimates highly depends on the starting year. Furthermore, the trends start to increase in 1997 (middle stratosphere) or 1998 (lower stratosphere), probably due to the turning point in ODS. The later the trend period starts after this turning point, the higher the trend estimate is, which has also been observed by Harris et al. (2015). Starting the trend for example in the year 2000, as it is done in other studies (e.g. Steinbrecht et al., 2017; Petropavlovskikh et al., 2018),

increases the GROMOS trend by almost 2 % per decade compared to the trend that starts in 1997. The trend dependency on the starting year is controversial to the definition of a linear trend, which does not change with time. This suggests that the true trend might not be linear, or that some interannual variations or anomalies are not captured by the trend model. Nevertheless, our findings demonstrate that it is important to consider the starting year and the trend period length when comparing trend estimates from different instruments or different studies.

## 4.5 Trend comparison

Trend profiles for all instruments at the three measurement stations are shown in Fig. 10. The trend profiles agree on a positive trend in the upper stratosphere, whereas they differ at lower altitudes. Due to the given uncertainties, most of the trend profiles are not significantly different from zero at 95 % confidence interval. Only GROMOS, the lidars and the ozonesondes at OHP

show significant trends in some parts of the stratosphere (bold lines in Fig. 10).

We observe that all instruments that measure in the upper stratosphere show a trend maximum between $\sim 4$ and $\sim 1.5\,\mathrm{hPa}$ (between $\sim 37\,\mathrm{km}$ and $45\,\mathrm{km}$), which indicates that ozone is recovering at these altitudes. The trend maxima range from $\sim 1$ to 3 % per decade, which is comparable with recent, mainly satellite-based ozone trends for northern mid-latitudes (e.g. Ball et al., 2017; Frith et al., 2017; Sofieva et al., 2017; Steinbrecht et al., 2017). Only the lidar trend at MOH is larger throughout the

whole stratosphere, with $\sim 3$ % per decade in the middle stratosphere and 4 to 5 % per decade between 5 and $2\,\mathrm{hPa}$. Nair et al. (2015) observed similar trend results for the MOH lidar, even if they consider five less years with a trend period ranging from 1997 to 2012. Steinbrecht et al. (2006) found that lidar data at MOH and OHP derive from reference satellite data above $35\,\mathrm{km}$ before 2003, with lower ozone at MOH and higher ozone at OHP compared to SAGE satellite data (Stratospheric Aerosol and Gas Experiment, McCormick et al. (1989)). Opposite drifts are reported by Eckert et al. (2014) after 2002 compared to

MIPAS satellite data (Michelson Interferometer for Passive Atmospheric Sounding, Fischer et al. (2008)). Combining those drifts might explain our high MOH trends and lower OHP trends. The distance of $\sim 600\,\mathrm{km}$ between the MOH and OHP stations might also explain some differences between the lidar trends. The GROMOS trend peaks at slightly lower altitudes than the trends of the other instruments. This difference might be related to the averaging kernels of GROMOS, which indicate that GROMOS retrieves information from higher altitudes than expected ($\sim 2\,\mathrm{km}$ difference).

In the middle and lower stratosphere, at altitudes below $5\,\mathrm{hPa}$ ($\approx 36\,\mathrm{km}$), our trends show discrepant results. The microwave radiometers and the MOH lidar report trends of 0 to 3 % per decade, and also the ozonesonde at OHP confirms this positive trend. However, the ozonesondes at MOH and Payerne as well as Aura/MLS and the OHP lidar report a trend of around 0 to $-2$ % per decade below $5\,\mathrm{hPa}$. Some of these observed trend differences can be explained by instrumental changes or differences in processing algorithms and instrument setup. The discrepancy between ozonesonde and lidar trends at OHP,





for example, are possibly due to the change of the pressure-temperature radiosonde manufacturer in 2007, which resulted in a step change in bias between ozonesonde and lidar observations. A thorough harmonization would be necessary to correct the trend for this change. The SOMORA trend shows a positive peak at 30 km, which is probably due to homogenization problems that are corrected in the new retrieval version of SOMORA (Maillard Barras et al., 2018). Furthermore, we have

shown that sampling rates and starting years have an important effect on the resulting trend. Our results show that the lidar sampling at MOH leads to a higher trend in the lower stratosphere than using a continuous sampling. The high lidar trend at MOH might therefore partly be explained by the sampling rate of the lidar. The trend period lengths differ between SOMORA, Aura/MLS and the other data sets, which might also partly explain differences in trend estimates. To explain the remaining trend differences in the lower stratosphere, further corrections e.g. for anomalies, instrumental changes or sampling rates would

be necessary. In brief, trends in the lower stratosphere are not yet clear. For some instruments, significant positive trends are reported, but for many other instruments trends are negative and mostly not significantly different from zero. This result reflects the currently ongoing discussion about lower stratospheric ozone trends (e.g. Ball et al., 2018; Chipperfield et al., 2018; Stone et al., 2018).

In summary, our ground-based instrument data agree that ozone is recovering around 3 hPa (40 km) with trends ranging

between 1 and 3 % per decade for most data sets. In the lower and middle stratosphere between 24 and 37 km (31 and 4 hPa), however, the trends disagree, suggesting that further research is needed to explain the differences between ground-based trends in the lower stratosphere.

## 5   Conclusions

Our study emphasizes that natural or instrumental anomalies in a time series affect ground-based stratospheric ozone trends.

We found that the ozone time series from the GROMOS radiometer (Bern, Switzerland) shows some unexplained anomalies. Accounting for these anomalies in the trend estimation can substantially improve the resulting trends. We further compared different ground-based ozone trend profiles and found an agreement on ozone recovery at around 40 km over Central Europe. At the same time, we observed trend differences ranging between −2 and 3 % per decade at lower altitudes.

We compared the GROMOS time series with data from other ground-based instruments in Central Europe and found that

they generally agree within ±10 %. Periods with larger discrepancies have been identified and confirmed to be anomalies in the GROMOS time series. We did not find the origins of these anomalies and assume that they are due to instrumental issues of GROMOS. The identified anomalies have been considered in the GROMOS trend estimations because they can distort the trend. By testing this approach first on a theoretical time series and then with the real GROMOS data, we have shown that identifying anomalies in a time series and considering them in the trend programme makes the resulting trend estimates more

accurate. With this method, we propose an approach of advanced trend analysis based on the work of von Clarmann et al. (2010) that may also be applied to other ground- or satellite-based data sets to obtain more consistent trend results.

Comparing the GROMOS trend with other ground-based trends in Central Europe suggests that ozone is recovering in the upper stratosphere between around 4 and 1.5 hPa (≈ 37 km and 45 km). This result confirms recent, mainly satellite-based



studies. At other altitudes, we have observed contrasting trend estimates. We have shown that the observed differences can partly be explained by different sampling rates and starting years. Other reasons might be nonconformity in measurement techniques, instrumental systems or processing approaches. Further, the spatial distance between some stations might explain some trend differences, because different air masses can be measured, especially in winter when polar air extends over parts

of Europe. Accounting for anomalies in the different data sets as proposed in the present study might be a first step to improve trend estimates. Combined with further corrections, e.g. for sampling rates or instrumental differences, this approach may help to decrease discrepancies between trend estimates from different instruments. In many other studies, the observed trend differences are less apparent because the ground-based data are averaged over latitudinal bands (e.g. WMO, 2014; Harris et al., 2015; Steinbrecht et al., 2017). Nevertheless, it is important to be aware that ground-based trend estimates differ considerably,

especially in the lower stratosphere. Exploring the origin of the differences and improving the trend profiles in a similar way as we did for GROMOS may be an important further step on the way to monitor the development of the ozone layer. To summarize, we have shown that anomalies in time series need to be considered when estimating trends. Our ground-based results confirm that ozone is recovering in the upper stratosphere above Central Europe, and emphasize the urgency to further investigate lower stratospheric ozone. The presented approach to improve trend estimates can help in this endeavour.

*Data availability.*  All ground-based data used in this study are available at ftp://ftp.cpc.ncep.noaa.gov/ndacc. The MLS data from the Aura satellite are available at http://avdc.gsfc.nasa.gov.

*Author contributions.*  L. Bernet and K. Hocke designed the concept and the methodology. L. Bernet carried out the analysis and prepared the manuscript. T. von Clarmann provided the trend model. All co-authors contributed to the manuscript preparation and the interpretation of the results.

*Competing interests.*  The authors declare that they have no competing interests. T. von Clarmann is co-editor of Atmospheric Chemistry and Physics but he is not involved in the evaluation of this paper.

*Acknowledgements.*  This study has been funded by the Swiss National Science Foundation, grant number 200021-165516.





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





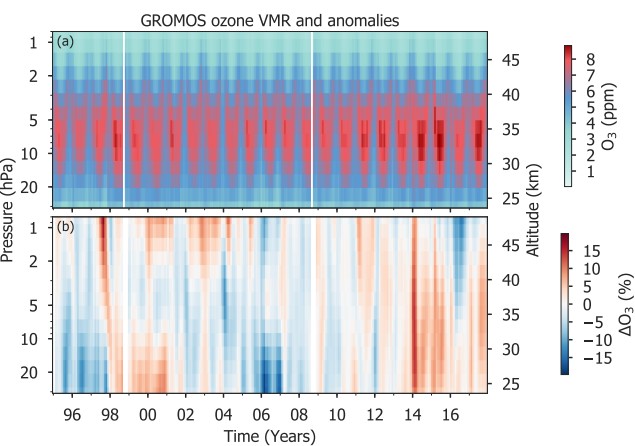

**Figure 1.** (a) Monthly means of ozone volume mixing ratio (VMR) measured by GROMOS (Ground-based Millimetre-wave Ozone Spectrometer) at Bern from Jan. 1995 to Dec. 2017. The white stripes indicate months with missing data. (b) Deviation from GROMOS multi-year monthly means (ozone climatology). The anomalies in (b) are smoothed by a moving average of 3 months.





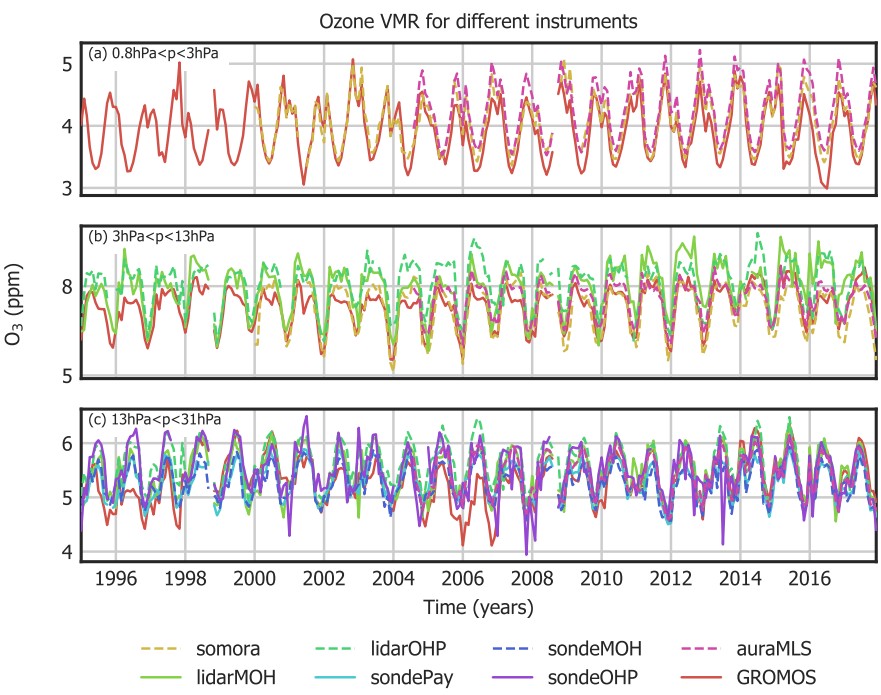

**Figure 2.** Monthly means of ozone VMR from the microwave radiometers GROMOS (Bern) and SOMORA (Payerne), the lidars at the observatories of Hohenpeissenberg (MOH) and Haute Provence (OHP), the ozonesonde measurements at MOH, OHP, and Payerne, as well as Aura/MLS data above Bern. The data have been averaged over three altitude ranges.





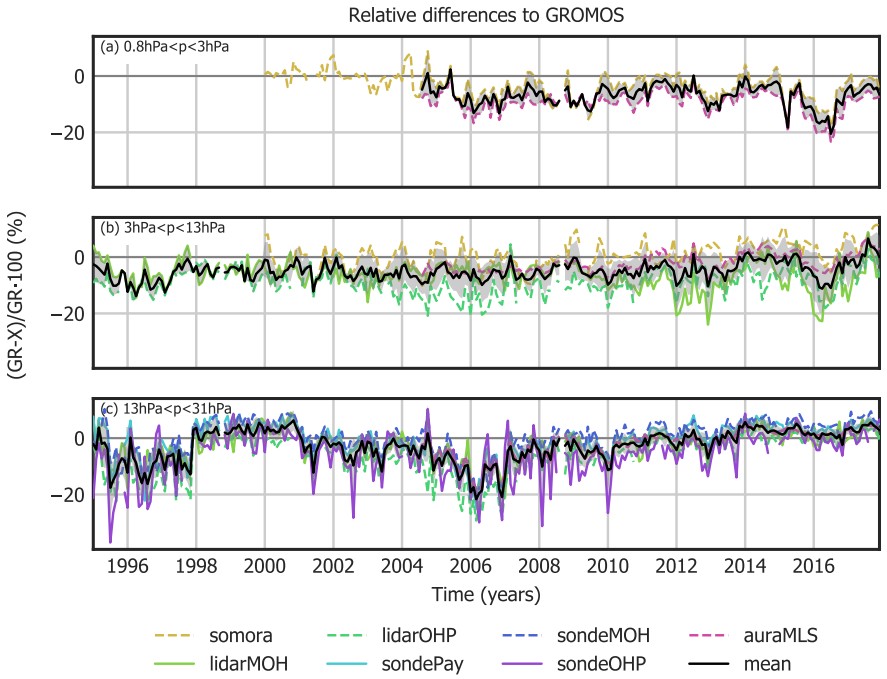

**Figure 3.** Monthly means of relative differences between GROMOS and coincident pairs of SOMORA, lidars (MOH, OHP), ozonesondes (Payerne, MOH, OHP) and Aura/MLS, averaged over three altitude ranges. The relative difference is calculated by $(\mathrm{GROMOS} - \mathrm{X})/\mathrm{GROMOS} \cdot 100$. The black lines show the mean values of all differences and the grey shaded areas shows its standard deviation.





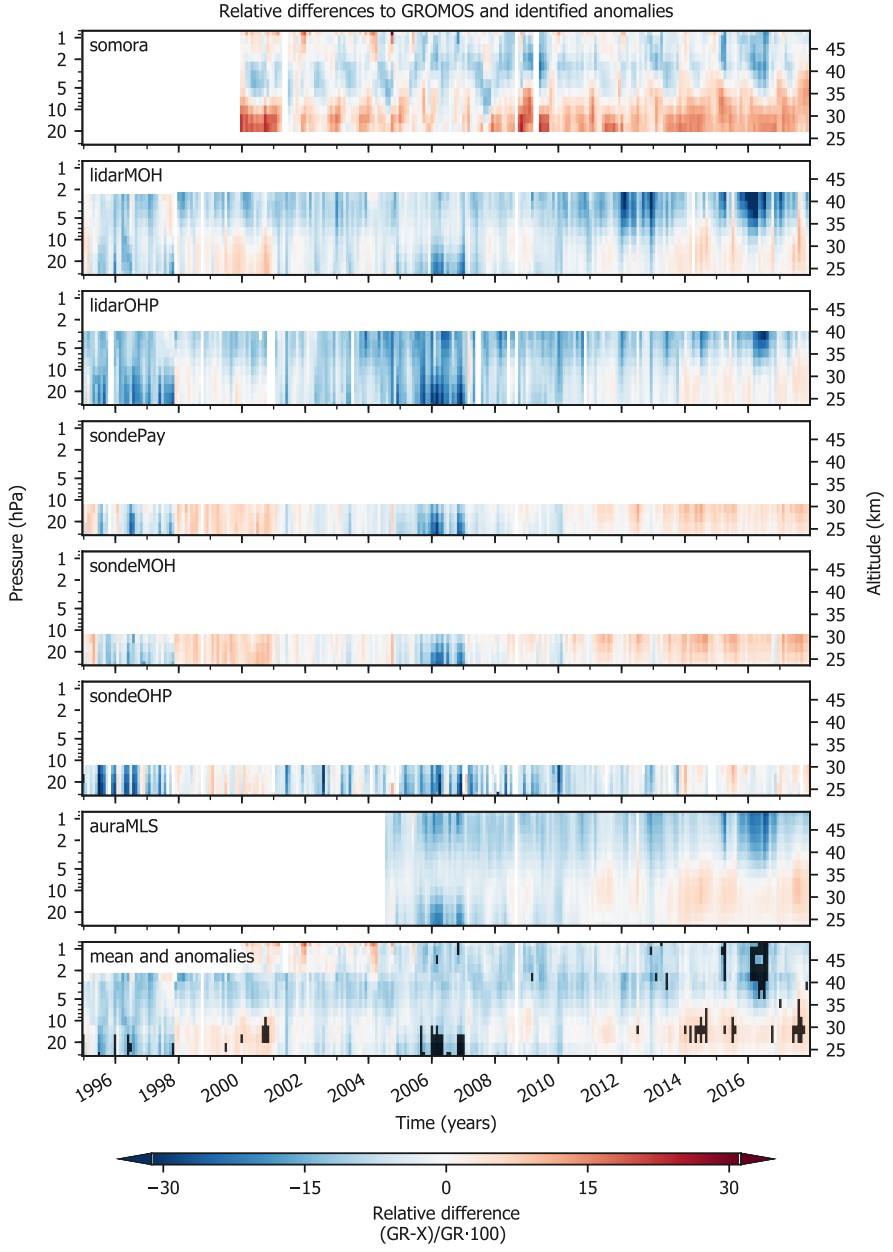

**Figure 4.** Monthly means of relative differences between GROMOS and SOMORA, lidars (MOH and OHP), ozonesondes (Payerne, MOH, OHP) and Aura/MLS. The lowest panel shows the mean of all relative differences. The black areas in the lowest panel show periods where at least three data sets (or two data sets above 2 hPa) have a relative bias of >10 %.





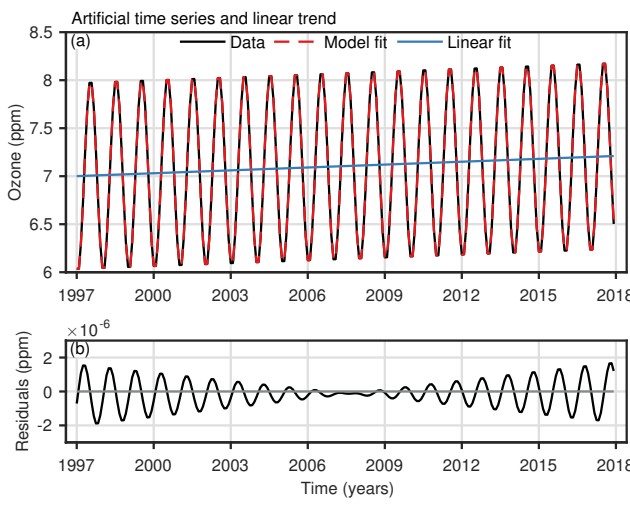

**Figure 5. (a)** Artificial ozone time series (composed of a simple seasonal cycle and a linear trend) and its model fit and linear trend estimation.
**(b)** Trend model residuals (data − model fit).





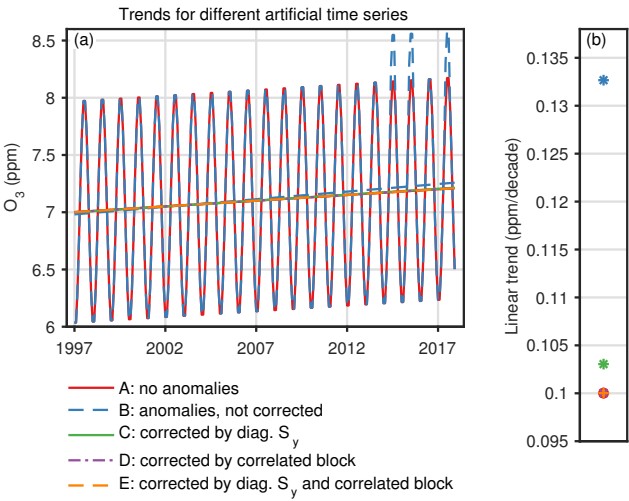

**Figure 6. (a)** Ozone time series and linear trends for the same artificial ozone time series as in Fig. 5 (case A). Anomalies were added to the time series in the summer months 2014, 2015 and 2017 in case B. Different corrections have been applied to account for those anomalies in the trend fit, represented by the cases C, D, and E. **(b)** Linear trend estimates for the time series without anomalies (case A) and the time series with added anomalies with different trend estimates, considering (cases C, D, E) or not considering (case B) the anomalies. For cases A, D, and E the trend is approximately the same ($\sim 0.1\,\mathrm{ppm}$) and therefore almost not discernible in panel (b).



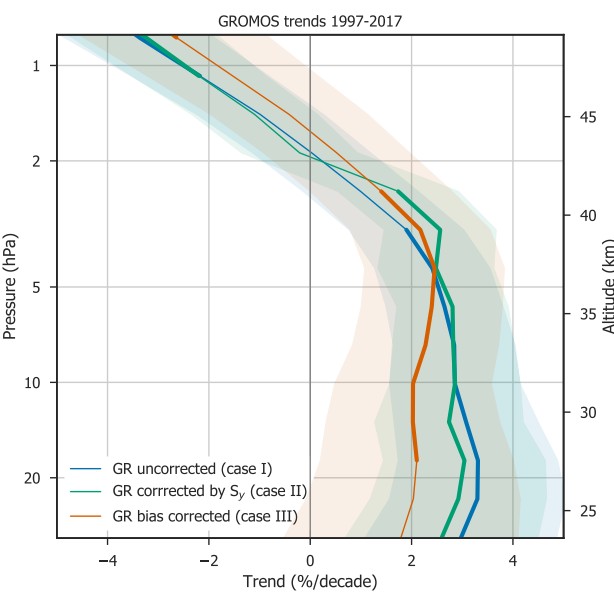

**Figure 7.** GROMOS ozone trends from Jan. 1997 to Dec. 2017, without considering anomalies (case I), considering anomalies in the diagonal elements of $\mathbf{S_y}$ ( case II), and considering a correlated bias block for anomalies (case III). The shaded areas show two standard deviations ($\sigma$), whereas the bold lines represent altitudes where the trend profile is significantly different from zero at 95 % confidence interval ($|\text{trend}| > 2 \cdot \sigma$).




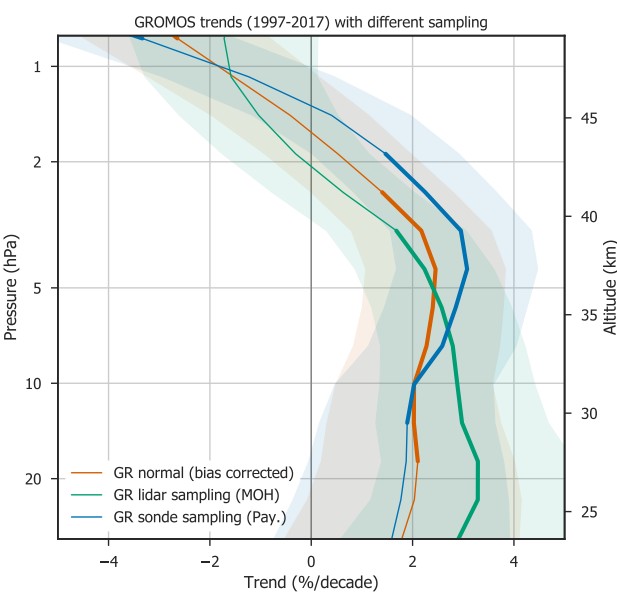

**Figure 8.** GROMOS trends from Jan. 1997 to Dec. 2017 using its high temporal resolution (normal data set with correction for anomalies), as well as the temporal sampling rate of the MOH lidar and the Payerne ozonesonde. The shaded areas show the 2-$\sigma$ uncertainty and bold lines identify trends that are significantly different from zero.



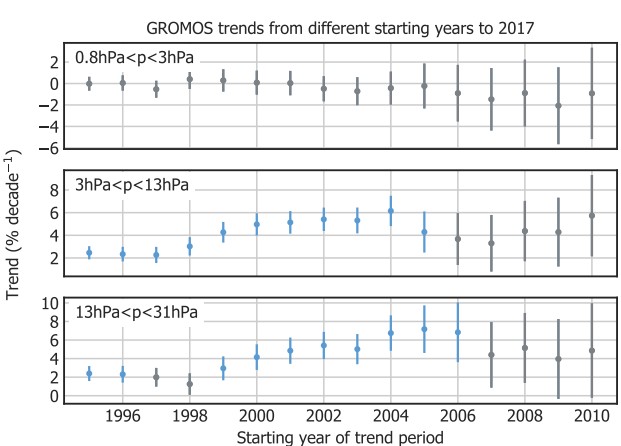

**Figure 9.** GROMOS trends averaged over three altitude ranges starting in different years, all ending in Dec. 2017. Trend estimates that are not significantly different from zero at 95 % confidence interval are shown in grey.



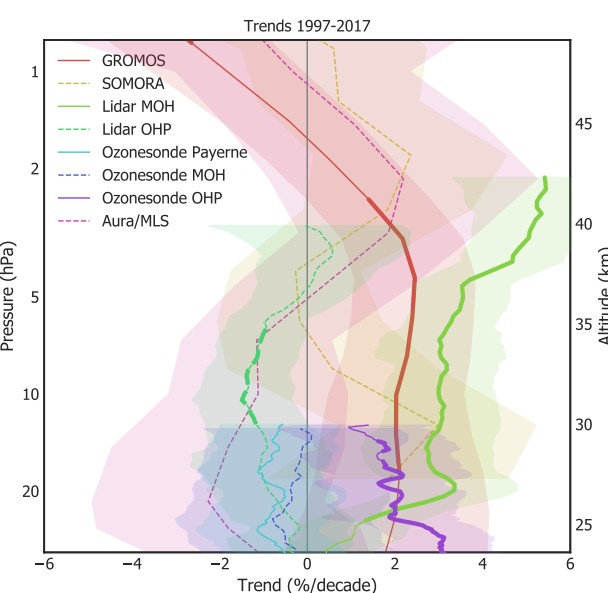

**Figure 10.** Ozone trends of different ground-based instruments in Central Europe and Aura/MLS (over Bern, Switzerland). The 2-$\sigma$ uncertainty is shown by shaded areas. Bold lines indicate trends that are significantly different from zero (at 95 % confidence interval).





**Table 1.** Information about measurement stations, instruments, and data used in the present study.

| Station | Instrument | Altitude range | Measurement rate | Analysis period |
|---|---|---|---|---|
| Bern, Switzerland 46.95 ° N, 7.44 ° E, 574 m | GROMOS | 31–0.8 hPa | 30 min to 1 h | 01/1995–12/2017 |
| Payerne, Switzerland 46.8° N, 7.0° E, 491 m | SOMORA Ozonesonde | 31–0.8 hPa 24–30 km | 30 min to 1 h 13 profiles/month[a] | 01/2000–12/2017 01/1995–12/2017 |
| Hohenpeissenberg, Germany 47.8° N, 11.0° E, 980 m | Lidar Ozonesonde | 24–42.3 km 24–30 km | 8 profiles/month[a] 10 profiles/month[a] | 01/1995–12/ 2017 01/1995–12/2017 |
| Haute Provence, France 43.9° N, 5.7° E, 650 m | Lidar Ozonesonde | 24–39.9 km 24–30 km | 11 profiles/month[a] 4 profiles/month[a] | 01/1995–12/2017 01/1995–12/2017 |
| Aura Satellite, above Bern 46.95±1° N, 7.44±8° E | MLS | 31–0.8 hPa | 2 overpasses/day | 08/2004–12/2017[b] |

[a] Averaged number of profiles per month in the analysed period.

[b] For the trend calculations, data from Jan.2005 to to Dec. 2017 are used.

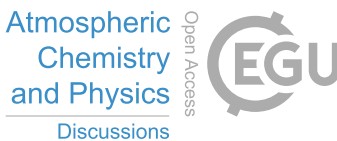

**Table 2.** Summary of the artificial ozone time series and the different model parameters used to correct the trend estimation for artificially added anomalies.

| | Parameters in the artificial time series | | | Parameters in the trend programme | | |
|---|---|---|---|---|---|---|
| Case | True trend | Monthly uncert.[a] | Added anomalies | Uncert. for anomalies[b] | Bias uncert.[c] | Estimated trend |
| A | 0.1 ppm | 0.1 ppm | – | – | – | $0.1 \pm 0.01$ ppm/decade |
| B | 0.1 ppm | 0.1 ppm | 5 % | – | – | $0.133 \pm 0.01$ ppm/decade |
| C | 0.1 ppm | 0.1 ppm | 5 % | 5 % | – | $0.103 \pm 0.011$ ppm/decade |
| D | 0.1 ppm | 0.1 ppm | 5 % | – | 5 ppm | $0.1 \pm 0.011$ ppm/decade |
| E | 0.1 ppm | 0.1 ppm | 5 % | 5 % | 5 ppm | $0.1 \pm 0.011$ ppm/decade |

[a] Uncertainty considered in the diagonal elements of the covariance matrix $\mathbf{S_y}$.

[b] Uncertainty considered in the diagonal elements of $\mathbf{S_y}$ for months with anomalies.

[c] Bias uncertainty considered in the off-diagonal elements of $\mathbf{S_y}$ for months with anomalies, set as a correlated block.