# Peer review of "Ozone trend profiles in the stratosphere: combining ground-based data over Central Europe to consider uncertainties"

_Atmospheric Chemistry and Physics, 2018_

## Referee Comment (RC1) · Anonymous Referee #1 · 1 Jan 2019

This is the first review of the paper titled "Ozone trend profiles in the stratosphere: combining ground-based data over Central Europe to consider uncertainties" by Leonie Bernet et al. The paper describes effect of the sampling and instrumental artifacts on derived trends. Paper focuses on the Microwave time series from Bern, Switzerland and investigates step changes in the record due to modification to the instrument in 2009. The impact of step changes in the record on the trend and uncertainties is tested with the synthetic data record. The temporal sampling and time period selected for trend analyses are also tested by authors to determine uncertainties in the derived stratospheric ozone trends. Authors compare other ozone records from several local and regional NDACC stations with Bern Microwave ozone record and find pos-

itive trends in the upper stratosphere, consistent between ground-based instruments and satellite records (Aura MLS record). Derived trends in the upper stratosphere are similar to previously published analysis (i.e. WMO Ozone assessment, 2014; Steinbrecht et al, 2017; Petropavlovskikh et al , 2018). Larger differences in derived trends are found in the middle and lower stratosphere. Authors conclude that their proposed anomaly tool will help to improve trend estimates for ground-based stations.

This is a very well written paper. Analyses of ground-based data records for Europe are done with great detail to instrumental artifacts, sampling biases and trend model assessment to assess impact of anomalies on the trends and uncertainties. Comparisons with satellite overpass data help to identify differences in the considered ground-based records and suggest steps to improve homogenization of the GB records. The detailed comments are provided below.

p. 1 abstract, line 9 . "We compare our improved GROMOS trend estimates with results DERIVED from other ground-based station OZONE RECORDS () in Central Europe." Add " derived and "ozone records". You might also indicate that these are NDACC records. p. 1 line 20 "Antarctic ozone started " -> "stratosphere ozone over Antarctica started " p. 2 line1 "Outside of the polar vortex, however, increasing ozone is more difficult to detect" – why? I always thought that due to large dynamical variability in the vortex position over Antarctic (or Arctic) it is more difficult to detect a trend as compared to less variability observed over other locations, especially in the upper stratosphere. Are you discussing low stratosphere trends here? p. 2 line 28 "All of these studies" – I does not feel right to use "these studies" at the beginning of the paragraph. May be you can change it to the "As discussed above, several published studies found similar in magnitude trends in the corrected satellite records . . ." p. 3 line 23, "altitude information of the ozone molecules" -> "information about vertical distribution of ozone molecules"? p. 4 line 2, "subtracting mean bias" – was it annual mean or monthly mean bias? p. 4, line 16 (or line 18) "3-days moving window" ( or 30-days moving window) – should it be "3-days moving average" ("30-days moving average")? p, 4, line 27 At the end of the

paragraph, please provide a brief description of main differences in the processing of these two station records ( i.e different spectroscopic datasets, averaging time period of the measurements, different error estimates, vertical resolution, smoothing, external information used in retrieval such as temperature profiles), some information that will let reader know if there are differences between the two retrieval methods such that they would create differences in the processed data originating from the same measurement. p. 5, line 1 "influence" -> interference? p. 5 line 2 "averaged measurement error" – is it derived for one night of averaging or for the entire month? Is it standard deviation of averaged profiles or error propagation analyses? p. 5 line 5 "host" sounds like lidar is temporary located in France. Would "operated" be a better way to describe it? p. 5 line 18. Instead of "those" use Dobson . Also, at the end sentence add "for normalization". p. 6 line 11 . "in the time period FROM Jan, 1995..." - add "from" p. 6, line 11."The different GB instruments..." – add GB (ground-based). May be you can abbreviate ground based earlier in the text. p. 6, line 27. "We therefore interpolate all lidar and ozonesonde profiles ..." Interpolation from altitude to pressure level requires knowledge of temperature . Temperature is typically available for ozonesondes from radiosonde measurement. But it is not always the case for the lidar (unless lidar also measures temperature). What temperature profiles is used to interpolate OHP and MOH lidar data from altitude to pressure level? Is it taken from reanalyses? ECMWF? p. 7, lines 26-27. "... was corrected by its mean relative difference to GROMOS" . Was the mean calculated over the entire period of the GROMOS/GB instrument time series? p. 7 line 29 "the respective month was identified as anomaly" Was anomaly identified in GROMOS data or in any of the compared instrumental records? p. 9 line 7 "the degrees of freedom of the trend fit" – can you please discuss this here – how you derive it. p. 9 line 21. "added to the corresponding part of covariance matrix" – can you provide more details? May be an example can be provided at this point to demonstrate the process? p. 9 line 33-34. "value of the corresponding dataset at each altitude" – did you already remove bias earlier in the process? And thus, this will be the same value for all datasets, yes? p. 11, line 15-17 "this can be explained by the low number of

OHP profiles..." – Why, if the two measurements sense the same ozone, there should be no large bias. Otherwise, what you are implying is that some large biases average out if one has a large number of coincident profiles.... I wonder if you should use coincident lidar measurements from OHP to evaluate if large biases between GROMOS and OHP sonds are indeed result of the spatial variability in ozone. Perhaps, you can plot PDF of subset of data (full or coincident) for both instruments to evaluate how they sample ozone variability over the same time-period. It will indicate if one instrument is not capable of obtaining low/high ozone, perhaps due to AK smoothing... p. 11, line 30-33. You indicated that positive anomalies in the middle and lower stratosphere in 2000 were confirmed through anomaly detection method. You also found positive bias in the upper stratosphere in 2000, although it was not confirmed by comparisons with other instruments (MLS and SOMORA). Would the middle and lower stratosphere biases suggests the instrumental anomaly in upper stratosphere in GROMOS as it seems to affect the entire profile? p. 12, line 21-22, why amplitude A is negative? Are coefficients in Eq (3) derived from observations? Why 0.1 ppm was selected for monthly uncertainty? Earlier you indicated that trend is 0.1 ppm per decade. What confidence level of the trend detection is expected in these time series? Also, please indicate the length of the artificial time series (yes, it can be deduced from Figure 5, but should be written in the first paragraph of section 4.1). p. 12, line 29. "($\sim$ 0.4 ppm) " - should it be 0.35 or 0.36 based on a=7 ppm in Eq (3) and the rate of change in time series is very small (0.1/120 x 12 (month per year) x 20 (years) = 0.2 ppm) p. 13, line 4, you provide uncertainty now as 5 %? What is it in ppm? Is it still 0.35 ppm? Then 0.1 ppm uncertainty for series A was 1.4 %? It is a bit confusing to switch between % and ppm. Also, it is not clear what "corresponding months" means. Do you mean that uncertainties for JJA months in 2014, 2017 and 2017 were increased to 5 % while the rest of month in the entire time series has 0.1 ppm uncertainty? p. 13, line 12-13 you state that trend estimate improves " the higher the uncertainties are chosen". Did you test model with larger than 5 % uncertainties? Why 5 % was selected in the first place? p. 14 line 11-15, "GROMOS trend peak is observed as slightly

lower altitude". " We observe a slight shift in the peak" of AK. – what do you mean? Can you explain the shift and when it occurred in time series? Or you mean that the maximum weight in AK profile for layer 35-40 km is higher by 2-4 km than the assumed peak of the AK at the middle of that layer? Then, you also suggest that the full width of the AK is between 20 and 25 km. If it is centered at $\sim$ 36.5 km, then you have to assume that the averaged (AK weighted) trend is estimated between 26.5 – 46.5 km altitude range. Can you please provide AK example for the layer? If you apply the AK smoothing to the lidar profile and calculate trends. Would these trends be different from trends derived by integrating ozone profile between 35 and 40 km? p. 14, line 16-17. The harmonization of time series before/after 2009 (application of bias and seasonal cycle correction) does not necessarily means that the entire bias in the retrieved profiles will be removed. Say, the change in the instrument optical characterization would produce a shift in the vertical distribution of the AK weights, the upgraded instrument would start sampling at higher/lower altitudes. And since ozone trends can be different at higher/lower altitudes, it would affect the homogenized time series, even after you remove the step-change bias. Can you show the AK before and after 2009 (for approximately the same atmospheric conditions)? p. 15, line 1. Can the temporal sampling test be done on the artificial time series (change in sampling from 1hour to 3 minutes)? p. 15, line 15-16. It is confusing to show "ozonesonde" results at altitude above 30 km. May be you should stop test at 30 km. p. 16, line 17 – is it 45 km or 42 km? p. 16, line 22 "derive" – this sentence seems to have some missing text. … p. 16, line 30 – change "our" to "derived" or "estimated". Also "discrepant results" means "differences in derived trends"? Figures 6 b – add uncertainty bars to the results. .You can also add circle slightly shifted horizontally to show where they fall. 7 and 8 – please consider to change either green or blue colors – they are too close to be easily distinguished in the plot. Also, please consider to change uncertainty envelops from solid color to shaded or not even filled envelops – it is hard to distinguish between different instruments. 10. – the same comment as above – selection of distinct colors and reduction in envelopes. May be the plot can be separate in two plots to reduce overlapping information – one

for MOH and one for OHP.

---

## Referee Comment (RC2) · Anonymous Referee #2 · 23 Jan 2019

**1  Short resume**

Outlying measurements are a nuisance to many types of analysis. Time series analysis in particular can be very sensitive to outliers, impacting both the estimated regression coefficients and their uncertainty. This topic is especially relevant for ozone profile trend assessments as the actual ozone trend is expected to be quite small (not more than $\sim 2\%$ per decade) which makes robust detection tricky and susceptible to a handful of outliers.

This paper starts off by presenting a comprehensive analysis of periods of outlying

measurements in the GROMOS time series. Where most other studies stop here and simply reject the outliers from further analysis, the authors apply a regression method proposed by von Clarmann et al (2010) which uses the timing of the detected/suspected outlying data and infers their magnitude along with other components in the regression model, such as the linear trend. Bernet et al. illustrate the value of the method using synthetic data, before applying it to the actual GROMOS time series. The methodology is explained well and the application is very relevant and adequately elaborated.

The last part of the paper moves slightly away from the core subject of the paper, and briefly discusses the impact of sampling and length of the time series on the GROMOS trend result. These are also compared to trends obtained from neighbouring ground-based data records and Aura MLS, however, the adopted outlier detection and incorporation method are not for these data sets.

**2  Recommendation**

This paper is very well written and the material presented supports the claims made by the authors. The subject is scientifically very relevant and fits the scope of ACP. I would therefore highly recommend to publish this paper as soon as my comments below have been incorporated.

**3  Major comments**

I ordered major comments in order of appearance in the text.

**3.1 Order of vertical regridding and smoothing**

p.7, l.7-9: The order of regridding and smoothing is unintuitive. First you downgrade highly resolved vertical profiles to a coarser grid, then convolve these with the MWR averaging kernel and a priori. In reality, the MWR instrument does not vertically smooth a coarse atmosphere. Shouldn't the procedure be the other way around? First smooth at best available vertical resolution, then regrid to the coarse grid. Hence you should oversample the averaging kernel in one direction to match the sampling resolution of the $x_h$, as in Eqs. 10-11 of Keppens et al. (2015, AMT).

**3.2 Unconventional choice of definition relative difference**

p.7, l.24: The relative difference is defined as $\Delta = 100 \times (GR - X)/GR$. Why did you not choose $\Delta' = 100 \times (GR - X)/X$, which is the more widely adopted definition? The ratio between both definitions is $\Delta/\Delta' = X/GR \geq 0$ (you removed all negative values; p.6, l.24-25). Your expression will be more sensitive to detect $GR < X$, the conventional expression is more sensitive to detect $GR > X$. Did you test the sensitivity of the detected periods to this definition?

**3.3 Description far away from application**

The description of the methodology (Sects. 2.2 and 2.3) is quite far away from the actual application. This makes it a slightly more difficult. You could consider explaining the outlier detection and regression methodologies right before their application in Sects. 3 and 4, rather than in Sect. 2.

**3.4 Outliers are only detected & considered for GROMOS**

You should state clearly that the proposed outlier detection and outlier incorporation methods are only applied to the analysis of the GROMOS time series. You could have done that for the other instruments as well, this would have resulted in a more fair comparison and improved the discussion in Sect. 4.5.

**3.5 Show trend results when outliers are removed**

Many, if not most, researchers remove outlying data points prior to regression. I am interested in seeing how this simple approach performs in the particular case of real GROMOS data (Sect. 4.2, Fig. 7) and of synthetic data (Sect. 4.1, Fig. 6). I acknowledge that this will not allow to draw conclusions that apply generally, since the result depends on the number and character of the outliers. At least this shows whether it was worth to invest extra coding time and analysis effort.

**3.6 Expand discussion to trend uncertainty**

The discussion in Sect. 4.1 is very insightful but does not treat another important factor in trend analysis: trend uncertainty. Can you elaborate by how much the outliers affect the trend uncertainty and whether the more advanced method is also able to provide more robust estimates of trend uncertainty?

One element that determines trend uncertainty is the noise in the time series. The example shown in Sect. 4.1 is hence an idealised case without random uncertainties. The few added outliers will therefore have a larger impact on the trend estimate than in the more realistic case where noise would have been present. Have you studied this effect? And please add a comment that the influence of outliers may be smaller in the case random errors would have been simulated as well.

**3.7 Positive trend at lower altitudes?**

Please tone down the statement on p.14, l.33-35: "This result implies that ozone might also recover at lower altitudes, but the uncertainties, the dependency on anomalies, and the insignificance of the corrected trend show that this result is less robust." I read it as (1) there is a recovery (2) but the result is less robust. So, do you claim a recovery in this altitude region, or not? In fact, it is the region where GROMOS trends differ most from those obtained by other satellite- and ground-based techniques. Hence, more caution is needed in how this is phrased.

**4 Minor comments**

p.1, l.0: The title does not convey the main message of the paper. It should be improved so readers will find your work more easily. First, the manuscript is all about detecting unusual observations and taking these into account in the regression. Hence the "to consider uncertainties" is too vague, I believe you should be more specific. Uncertainties can mean anything, really. Second, the presence of "combining ground-based data" suggests that you merged data sets prior to regression, which is not the case. This is misleading, which is definitely undesired for a title of a paper. What about something along the lines of "Combined study of ground-based ozone profile data over Central Europe to detect and incorporate anomalous observations in the analysis of trends in the stratosphere"

p.1, l.1-15: The abstract should be improved, as it is not clear exactly what the paper adds to the existing body of literature. For instance, "[...] an approach for handling suspicious anomalies [...]" does not say how you handle anomalies. This can be done in many ways. In my view, you should state clearly (1) that you analysed multiple ground-based data records to detect anomalous periods in the Bern MWR data record,

and (2) that you then used this information to regress the Bern time series for long-term trends and the temporary bias for the detected anomalous data taking periods.

p.1, l.18: "[...] in recent years [...]". Add LOTUS Report and replace (WMO, 2014) by (WMO, 2018) to the citation. Also, in the submitted version you sometimes refer to (WMO, 2014) and sometimes to (Pawson et al, 2014). Choose just one of these where relevant.

p.2, l.6-8: Incorrect language. The linear trend fitted represents the linear component in the time series after all other natural variations are accounted for (solar, QBO, ENSO, aerosol, ...). This linear term is expected to receive contributions from decreasing ODS and increasing GHG, and possibly other unidentified processes. It is not possible to fit a linear term due to ODS and another linear term due to GHG, since these terms are fully dependent. Hence, it is incorrect to state that "a trend might be masked by natural variability and factors such as GHG increases and changing dynamics". So this needs to be rephrased.

p.2, l.10: Different in what sense? If you combine data records from different instruments, ok. But here such a merging is not considered. So a "changing resolution" or a "changing sampling pattern" is more relevant for a time series composed of measurements by one instrument.

p.2, l.11: Add Frith et al. (2017) to the citation. The paper is more recent but also deals with ozone profiles which is closer to what you report in the manuscript.

p.2, l.12-14: The influence of outliers at start and end points on the trend estimates is well known and not discovered by Bai et al as suggested by the current phrasing. I propose to move the citation to the end of the phrase such as (e.g., Bai et al., 2017).

p.2, l.20: I am not sure there is evidence that all data sets used by Steinbrecht et al. (2017) have been successfully corrected for drift. Hubert et al. reported drift in parts of the HALOE, SCIAMACHY and GOMOS data sets. These have not been corrected.

And so far no one has proven that the OSIRIS correction is removing the entire drift. Please rephrase this sentence.

p.2, l.34: Add a reference to the LOTUS report as it contains the most recent and complete set of trend estimates.

p.3, l.1: Confusing language "handle [...] anomalies by considering anomalies [... ]". It would help to define up front what you mean with "anomaly". Is it due to the instrument, due to natural processes, ...? A great deal of readers will read anomaly as something that is not problematic, as it is caused by natural processes. Others will interpret it as an issue to solve.

p.3, l.5: "[...] resulting in a corrected trend estimate [...]". Corrected with respect to what? Perhaps "improved" is more in place here?

p.3, l.12: The word "anomaly" is often also used for anomalous geophysical events. This class of anomalies is not the subject of this manuscript. In caption of Fig. 1 you use "anomaly" in the sense of a "deviation from a climatology", which agrees with how many readers will interpret it. To avoid any confusion I suggest to define "anomaly" unambiguously at the beginning of the manuscript (and perhaps repeated here and there in the text). And are inhomogeneity (see comment below) and anomaly the same for you?

p.3, l.11: Section 2 is a tad long. If you consider trimming the manuscript (which is not what I propose), then this section would be a good place.

p.4, l.16: Add "mean" or "median" to the moving window to clarify what statistical measure you used for the location of the smoothed data.

p.5, l.17: Replace by "[...] using concurrent total column ozone data from the Dobson [...]". This to clarify that the corrections are done using data taken at the same/similar time.

p.6, l.4: You mentioned the spectral line for the ground-based MWR, so it would be

nice to have the frequency of the line for Aura MLS as well.

p.6, l.7: Improve language "stable with insignificant drifts", this is twice the same information. Either use "stable", or more precisely "stable within X% per decade", or just "because there are no drifts between".

p.6, l.9: Clarify what data is compared. E.g. "Comparison of time series". (Further on in the paper there is also a comparison of the trends.)

p.6, l.17: Why is the GROMOS retrieval accurate between 31-0.8 hPa? The retrieval is not accurate simply because the measurement response is high. The instrument can still be biased (large systematic error) and even imprecise (large random uncertainty). The use of "accurate" suggests a small uncertainty. Is that really the case?

p.6, l.22-24: The native measurement coordinate system for lidar is number density versus altitude. You need to explain how the lidar data are converted as well. For ozonesonde this is trivial due to the temperature data from the attached radiosonde.

p.6, l.26-29: Please clarify the details of the regridding procedure. The description is confusing and I am not sure I understand the order of operations and on what observed quantities.

p.6, l.26-29: Also, is the interpolation a simple linear interpolation or is it a more advanced method such as described in Calisesi et al. (2005)?

p.7, l.3 and p.7, l.21: Remove "direct" from "direct comparison", since there are no indirect comparisons in this manuscript.

p.7, l.25-27: The use of "corrected" relative difference is misleading. Why not e.g. "debiased" relative difference or "variations of the" relative difference? You simply remove the mean level to look at deviations of the relative difference w.r.t. its mean level. Please abstain from using "corrected relative difference" elsewhere in the text as well.

p.7, l.25-27: Add formula to clarify that you remove a constant offset in $RD$. In other

words, assuming $RD_i$ is the monthly mean of month $i$ and the mean of $RD$ over all months is $\mu_{RD}$, then you debiased the $RD$ time series using $RD_i - \mu_{RD}$.

p.8, l.19-20: Von Clarmann present an approach to consider "known", "identified" or "suspected" inhomogeneities in the trend analysis. Their method does not identify the time of an inhomogeneity, but co-fits the magnitude of the temporal offset. Second remark, state right here that you apply the von Clarmann method and code.

p.8, l.15-...: Now you start to use "inhomogeneities" rather than "anomalies". Is there a difference between these terms? Please define these clearly, if they are. If not, then state that as well.

p.8., l.15: Strengthen the link with previous Section 2.2. State asap that you will now use the times of the anomalies detected by the previously described method (Sect. 2.2) in the regression scheme (Sect. 2.3).

p.8, l.15-16 and/or l.23-24: Clarify that the full error covariance matrix describes the correlation between measurements, not between pressure levels.

p.8, l.21-22: Remove the entire phrase "Another reason for [...]". Instead add this possible cause (unknown origin) to the end of the phrase on l.16-18 "Inhomogeneous data can originate from [...] spatial or temporal sampling." (which currently lists known causes). The method you described in Sect 2.2 identifies periods of anomalies, but does not identify the cause of the anomaly. It can be anything in the list of l.16-18, but also unknown instrumental issues.

p.9, l.5: "no error correlation", but in what domain? Clarify that you deal with the correlation between measurements, not between pressure levels.

p.9, l.5: You introduce a second VCM (variance-covariance matrix). It would help to have a separate symbol for the different VCM's ($S_1$, $S_2$, $S_{instr}$, $S_{autocorr}$ or whatever) and use in-line formulas to explain when you add these and when not. This allows you to get away from unclear language such as "additional covariance matrix" and

"enhanced covariance matrix". You could just refer to it as $S' = S_1 + S_2$. This also

p.9, l.12: Clarify "which means the time correlated residuals" are not considered ($S_2$ or $S_{autocorr}$).

p.9, l.15: Now you switch back to using "anomalies", not "inhomogeneities". Be consistent or mention that both terms have identical meaning.

p.9, l.16: It is important to link with previous section. "[...] for months where anomalies were identified (using the method described in Sect. 2.2)".

p.9, l.18: Are you using the so-called "corrected relative differences" or not?

p.9, l.24: Replace "low" values by "small" values. E.g. -25% is a very low value, but not a small value when compared to -5%. I know the example is not applicable here, but I would generally suggest to refrain from using "low/high" and use "small/large" instead when referring to the magnitude of values (not the sign).

p.9, l.29: Clearly state that above procedure is only applied to GROMOS data. You did not consider outliers in any other time series than GROMOS.

p.9, l.32-34: The percentage trend is obtained by dividing the ppmv trend by the mean level. If there is a trend the mean value will depend on the period over which the mean is computed. Did you use exactly same period for all trends? Since Aura/MLS has the shortest record, I would guess the ppmv trend should be scaled to 2005-2017 mean values. Has this been done? If not, please state whether this effect is negligible.

p.10, l.1: You could remove ", as described by Tiao et al. (1990)", as this is just introductory-level Gaussian statistics.

p.11, l.1-2: You refer to "another temporal sampling", another than the other instruments? Another than the usual GROMOS sampling?

p.11, l.9-10: Remove ambiguity. You deal with relative differences of monthly mean values of time coincident pairs, not of the monthly mean value of the relative difference

of time coincident pairs. Figure 3 suggests the latter however.

p.11, l.15-17: The large outliers in the OHP comparison time series (Fig. 3) are all negative. The explanation is not satisfying, if not incorrect. If outliers are due to the small sample, then large positive peaks should be observed as well. Unless there is a systematic component in both the sampling pattern and the ozone fields. The latter seems highly unlikely to me. Can't it, instead, be due to the presence of large positive outliers in the OHP ozonesonde record?

p.11, l.9: Does Fig. 3 show the "corrected relative difference"? I guess not, since the mean level seems to differ from zero. This should be clarified.

p.11, l.21: Doesn't Fig. 4 show the "corrected relative difference" (or debiased relative difference, see earlier comment above)? Mention this clearly in the text, and abstain from using "corrected relative difference", I like "debiased relative difference" better.

p.11, l.34: Replace by "[...] comparing GROMOS to SOMORA [....]". There is only one reference instrument in the US during the period before Aura/MLS.

p.12, l.1: Add "[...] at these altitudes, especially prior to the start of the MLS measurements in 2004." And you could then e.g. also mention the step change around 2005 visible in panel (a) around 2005 which may be related to the change of SOMORA front end (see also comment below; p.28, Fig.3).

p.12, l-7.8: It is not entirely clear whether you are saying here that you disprove the claim by Steinbrecht (2009)? If yes, then end the phrase by removing the ambiguity. For instance: "[...] due to instrumental issues of GROMOS and not due to sampling."

p.12, l.7-8: Are there any leads on what could be the cause of the instrumental issues? If yes, it may be of interest to the reader.

p.12, l.21: A negative amplitude has no meaning. Is this a typo?

p.12, l.26: Vague language "This might be due to some error propagation". This clause

does not hold additional information and may as well be removed.

p.13, l.4: You arbitrarily chose 5%, which is $3.5 = 5/(0.1/7)$ times larger than $1.4\% = 0.1/7$. What if you increase to even higher uncertainties, e.g. in the limit of infinite uncertainty? This case would be equivalent to removing the anomaly from the time series prior to regression. (See also my major comment)

p.13, l.6: Text is inconsistent with caption of Fig. 6. Shouldn't it be "[...] once without (case D) and once with (case E) the increased [...]"?

p.13, l.7: Add to the last phrase that a 5 ppm correlated block essentially leads to a free fit of the magnitude bias.

p.13, l.27: Repeat by how much you increased the uncertainty, so a reader does not have to go back all the way to Sect 2.3.

p.13, l.34: "[...] decreases the trend slightly in the lower stratosphere, but the differences are small." Doesn't this point to a too small scaling of the uncertainties?

p.14, l.4: I still don't like the phrasing "corrected" trend profile. Corrected w.r.t. what? See also my comment p.3, l.5.

p.14, l.4 and l.30-31: "The corrected trend profile agrees well with recent satellite-based ozone trends [...]". I am not convinced that the case III results agree well with other studies. They are consistent with Steinbrecht (2017; Fig.5, Table 6) between 10–2 hPa (what you call "middle" stratosphere). However, there is tension between the results at 30-10 hPa ("lower" stratosphere). The other studies find smaller trend values, more closely to 0–1% per decade.

p.14, l.13-14: Somewhat confusing phrasing "a slight shift in the peak height". A shift with respect to what? Is the peak shifting in time, which is what I first think of in the context of time series analysis? More likely you mean that the peak sensitivity of the GROMOS retrievals is not at the true altitude?

p.14, l.30-31: See earlier comment, I would refrain from using "agree". The results are consistent, but do not agree well especially below the 10 hPa level. The bias corrected trend does lead to a better agreement

p.15, l.16: Unlike Reviewer 1, I like the fact that the full altitude range is shown. This section is not about a comparison to actual ozonesonde or lidar trends, but about the impact of sampling on GROMOS trends. I would keep the altitude range in Fig. 8 as is.

p.15, l.17-19: This is a very nice piece of information. Out of curiosity, have you tested to randomly subsample the GROMOS time series to $N_{MOH}$ and $N_{Pay}$, then regress? Such an analysis would add information about the random nature of the impact of sampling. The more positive sonde trend at 3 hPa may as well turn into a less positive trend if another temporal GROMOS sampling was picked but with same sample size as the ozonesonde time series. Would be nice, but not necessary.

p.15, l.26: Replace "estimations" by "regression".

p.15, l.26-27: Did you average the GROMOS trends or the GROMOS time series? If the former, how did you do that and does the trend of vertically averaged data agree with the vertical average of the trends?

p.16, l.5-6: Vague language "The trend dependency [...] is controversial to [...]". Please rephrase, I do not understand what you mean.

p.16, l.6-7: "This suggests that the true trend might not be linear, or that some interannual variations or anomalies are not captured by the trend model" This is known for a long time. Please rephrase to "This illustrates that [...]".

p.17, l.2-3: "A thorough harmonization would be necessary to correct the trend for this change." In fact, your manuscript describes a method that does not require harmonisation. I understand the extra work effort, but it is a pity for Section 4.5 that you did not treat outlying data in the non-GROMOS regression analyses. This would have made an even more interesting comparison.

p.17, l.4: It is unclear whether you have used these newer SOMORA retrievals or not. Please clarify in the revised manuscript.

p.26, Fig.1: "[...] indicate months without data." Missing data does not imply there are no data, and you mean there are no (screened) data during these months.

p.26, Fig.1: Mention the period over which the climatology is computed. So, "Deviation from GROMOS monthly mean 1997-2017 climatology" would make a better description.

p.28, Fig.3: "Monthly means of relative differences" implies you compute relative difference $100 \times (GR - X)/GR$ for each coincident pair, then take the monthly average. This is not how you described it in Sect. 2.2.

p.28, Fig.3: Do these time series represent the "relative difference" or the "corrected relative difference"? If these are corrected relative differences, then how can it be that most curves seem to have a negative mean value where zero would be expected?

p.28; Fig.3: Panel (a) shows a step around 2004–2006. Is this due to the change of front end in SOMORA in 2005 (p.4, l.21)? If yes, then it would be nice to at least mention this in the last paragraph of p.11.

p.29, Fig.4: Do all panels represent the "relative difference" or the "corrected relative difference"? Clarify in the caption in the latter case.

p.34, Fig.9: See my earlier comment. Did you average the trend, or average the time series then derive the trend? In the first case, how exactly did you compute the error on averaged trend?

p.34, Fig.9: Mention significance level of the error bars. I assume that these represent $11\sigma$ since some bars that do not cross the zero level are greyed out.

p.37, Table.2: Add time unit in first column. It should be "ppm/decade".

p.37, Table.2: Move third column ("Monthly uncert.") to first column in section "Parameters in the trend programme". No uncertainty has been added in the time series.

p.37, Table.2: Use consistent number of significant digits. I suggest three digits for all numbers quoted.

**5  Technical corrections**

p.1, l.4-5: Replace by "[...] anomalies that may all mask a [...]".

p.1, l.7: Remove "[...] to improve the derived trend profiles [...]". This is evident.

p.1, l.11: Replace by "[...] in agreement with satellite [...]".

p.1, l.17: What is a "serious" decrease? Perhaps "fast" or "large" is more appropriate language?

p.1, l.20: Replace by "[...] Antarctic ozone concentrations started [...]".

p.2, l.1: Replace by "[...] increasing ozone levels are more difficult [...]".

p.2, l.1: Replace by "[...] to detect and these depend on altitude [...]".

p.2, l.7: Add "s" to "[...] greenhouse gas increases [...]".

p.2, l.19-20: Replace by "[...] with no or small drifts [...]".

p.2, l.25: Remove "data" from "[...] data steps [...]".

p.2, l.27: Move the "yet" to "[...] has not yet been [...]".

p.3, l.21: Replace by "[...]  They measure the 142 GHz line where ozone molecules [...]". Currently you have "[..] atmospheric emission [...] molecules emit [...]".

p.3, l.29: Replace by "[...] is the main focus [...]".

p.4, l.10: Replace "instead" by "rather than from".
p.4, l.12: "data corrections".

p.4, l.13: "troposphere" is possibly more precise terminology here, rather than "atmosphere"? This then also links better with next phrase which contains "tropospheric humidity".

p.4, l.16: "[...] 3-day moving [...]". Drop "s".

p.4, l.18: Replace by "[...] outliers exceeding four times [...]".

p.4, l.18: "[...] 30-day moving [...]". Drop "s".

p.4, l.34: Move the last phrase "The lidar can only [...] the influence of sunlight." to the first paragraph of Sect. 2.1.2. This to avoid confusion that this information holds for all lidars, not just the one installed at Hohenpeissenberg which is the subject of this paragraph.

p.5, l.9: Remove newline, and ensure the phrase "More detailed information [...]" is next to the one ending in "[...] with the GROMOS limits." Since the phrase only contains information for OHP.

p.5, l.28: Replace by "[...] 30 km, above which the balloon [...]". The balloon does not usually burst at "30 km", but at "30 km or above".

p.6, l.2: Replace by "[...] Aura satellite, launched in mid 2004, [...]".

p.7, l.4: Replace by "The vertical resolution of GROMOS and SOMORA are usually coarser than for the other instruments. [...]".

p.7, l.17: Replace by "[...] GROMOS and SOMORA have a higher temporal resolution than the other instruments. [...]".

p.8, l.18: "temporally correlated".

p.8, l.22: Remove "that lead to anomalies". Otherwise "[...] reason for inhomogeneities might be [...] issues that lead to anomalies [...]".

p.9, l.7: Replace "adjusted" by "scaled", because that is what you do.

p.9, l.8: Replace by "[...] the trend fit becomes unity."

p.9, l.20: Some irrelevant information. Replace by "[...] In a second step, we account for biases [...]".

p.9, l.27: Drop "have" from "we have found".

p.9, l.31: Remove "and the Aura/MLS trend starts in Jan. 2005", as this is duplicated in the next phrase.

p.9, l.32: Replace by "Aura/MLS cover the shortest trend period, starting in [...]".

p.9, l.33: Add "regression" to "regression model output", to avoid any confusion with e.g. CCM's.

p.10, l.28: Replace altitude "classes" by "ranges".

p.11, l.1: "[...] whereas the GROMOS data shown here [...]".

p.12, l.20: Replace by "[...] per decade, i.e. $b = 0.1$ [...]".

p.12, l.25-26: Replace by "The residuals are of order $10^{-6}$ and increase towards the start and end of the time series (Fig. 5(b))."

p.13, l.12-13: Rephrase, it is hard to read. "We further found that the trend estimate is closer to its true value when higher uncertainties are chosen [...]".

p.14, l.11: Replace by "[...] even though the GROMOS [...]".

p.16, l.22: How do lidar data "derive" from satellite data? Did you mean "deviate", perhaps?

p.17, l.29: Replace by "[...] in the trend analysis [...]".

p.26, Fig. 1: Not "stripes" but "lines".

p.32, Fig.7: Remove the space before "case II" in the parenthesis on the second line of the caption.

---

## Author Comment (AC1) · 13 Mar 2019

**Author's response**

**Ozone trend profiles in the stratosphere: combining ground-based data over Central Europe to consider uncertainties**

Bernet et al., Atmos. Chem. Phys. Discuss., 2018, https://doi.org/10.5194/acp-2018-1213

Dear Referees, dear Editor Jens-Uwe Grooß,

we would like to thank the referees for the detailed investigation of the manuscript and for the constructive and elaborated comments and suggestions. We have taken the remarks into account and present a detailed answer in the following.

According to the Editor's comment, we repeated our analysis without excluding negative values, to avoid the appearance of an artificial positive bias. The text and all figures have been adapted accordingly.

We attach a revised version of the manuscript with marked changes and hope that we have responded satisfactorily to the suggestions and remarks. The referee's comments are given in italic typeface, our responses are given in blue, and the corresponding changes in the manuscript in grey.

Kind regards,
Leonie Bernet (on behalf of all co-authors)

**Contents**

**1 Author's response to Referee #1**

**1.1 General comment**

*This is the first review of the paper titled "Ozone trend profiles in the stratosphere: combining ground-based data over Central Europe to consider uncertainties" by Leonie Bernet et al. The paper describes effect of the sampling and instrumental artifacts on derived trends. Paper focuses on the Microwave time series from Bern, Switzerland and investigates step changes in the record due to modification to the instrument in 2009. The impact of step changes in the record on the trend and uncertainties is tested with the synthetic data record. The temporal sampling and time period selected for trend analyses are also tested by authors to determine uncertainties in the derived stratospheric ozone trends. Authors compare other ozone records from several local and regional NDACC stations with Bern Microwave ozone record and find positive trends in the upper stratosphere, consistent between ground-based instruments and satellite records (Aura MLS record). Derived trends in the upper stratosphere are similar to previously published analysis (i.e. WMO Ozone assessment, 2014; Steinbrecht et al, 2017; Petropavlovskikh et al , 2018). Larger differences in derived trends are found in the middle and lower stratosphere. Authors conclude that their proposed anomaly tool will help to improve trend estimates for ground-based stations.*

*This is a very well written paper. Analyses of ground-based data records for Europe are done with great detail to instrumental artifacts, sampling biases and trend model assessment to assess impact of anomalies on the trends and uncertainties. Comparisons with satellite overpass data help to identify differences in the considered ground-based records and suggest steps to improve homogenization of the GB records. The detailed comments are provided below.*

We thank the referee for the positive feedback and give detailed answers to the comments in the following.

**1.2 Specific comments**

*p.1 abstract, line 9 . "We compare our improved GROMOS trend estimates with results DERIVED from other ground-based station OZONE RECORDS () in Central Europe." Add " derived and "ozone records". You might also indicate that these are NDACC records.*

Corrected.

*p.1 line 20 "Antarctic ozone started" -> "stratosphere ozone over Antarctica started "*

Corrected.

*p.2 line1 "Outside of the polar vortex, however, increasing ozone is more difficult to detect" - why? I always thought that due to large dynamical variability in the vortex position over Antarctic (or Arctic) it is more difficult to detect a trend as compared to less variability observed over other locations, especially in the upper stratosphere. Are you discussing low stratosphere trends here?*

The text was adapted to follow your comment.

Outside of the polar regions, however, differences in ozone recovery are observed depending on altitude

and latitude.

*p.2 line 28 "All of these studies" - I does not feel right to use "these studies" at the beginning of the paragraph. May be you can change it to the "As discussed above, several published studies found similar in magnitude trends in the corrected satellite records . . ."*

The above presented studies agree on positive ozone trends in the upper stratosphere with some differences in magnitude, [..]

*p.3 line 23, "altitude information of the ozone molecules" -> "information about vertical distribution of ozone molecules"?*

Corrected.

*p.4 line 2, "subtracting mean bias" - was it annual mean or monthly mean bias?*

It was the mean over the whole overlapping period. We adapted the sentence to clarify.

[...] by subtracting the mean bias profile averaged over the whole overlap period ($FB_{mean} - FFTS_{mean}$) from all FB profiles (Moreira et al., 2015).

*p.4, line 16 (or line 18) "3-days moving window" ( or 30-days moving window) - should it be "3-days moving average" ("30-days moving average")?*

It has been corrected to "moving median window", which is more precise than "average", because the median has been used as statistical average in the smoothing window (see also comment from Ref. #2, p.4, l.16).

*p.4, line 27 At the end of the paragraph, please provide a brief description of main differences in the processing of these two station records ( i.e different spectroscopic datasets, averaging time period of the measurements, different error estimates, vertical resolution, smoothing, external information used in retrieval such as temperature profiles), some information that will let reader know if there are differences between the two retrieval methods such that they would create differences in the processed data originating from the same measurement.*

There are indeed differences in the retrievals of GROMOS and SOMORA instruments, mainly because the retrievals are optimised for each instrument. We agree that it would be a good idea to compare in more detail the processing differences of both instruments. However, this would also require a deep analysis of how the different parameters affect the ozone data to draw conclusions about the differences in ozone profiles, which would be beyond the scope of the present study. We therefore refrain from adding more detailed information about the processing in the manuscript and think that the description in Sect. 2.1. fulfils the requirements for the further analysis in our study. For more information regarding the retrieval, please refer to Moreira et al. (2015) for GROMOS and Maillard Barras et al. (2009) and Maillard Barras et al. (2015) for SOMORA.

*p.5, line 1 "influence" -> interference?*

Corrected.

The lidars can only retrieve ozone profiles during clear-sky nights due to scattering on cloud particles and the interference with sunlight.

*p.5 line 2 "averaged measurement error" - is it derived for one night of averaging or for the entire month? Is it standard deviation of averaged profiles or error propagation analyses?*

The "averaged measurement error" mentioned here is the mean value of all uncertainty profiles given in the study period. For each individual lidar ozone profile averaged over one night, the uncertainty is given by the statistical error of the signal. We adapted the text to clarify this.

In this study we limit the data to the altitude range where the measurement error averaged over the whole study period is below 10 % [...].

*p.5 line 5 "host" sounds like lidar is temporary located in France. Would "operated" be a better way to describe it?*

Corrected.

*p.5 line 18. Instead of "those" use Dobson . Also, at the end sentence add "for normalization".*

Corrected.

*p.6 line 11 . "in the time period FROM Jan, 1995. . ." - add "from"*

Corrected.

*p.6, line 11."The different GB instruments. . ." - add GB (ground-based). May be you can abbreviate ground based earlier in the text.*

We adapted the sentence after changing the structure of the methodology parts (see major comment 3 from Referee #2).

The stratospheric ozone profile data used in the present study comes from different ground-based instruments that measure in Central Europe (France, Germany, and Switzerland, Table 1).

*p.6, line 27. "We therefore interpolate all lidar and ozonesonde profiles . . ." Interpolation from altitude to pressure level requires knowledge of temperature . Temperature is typically available for ozonesondes from radiosonde measurement. But it is not always the case for the lidar (unless lidar also measures temperature). What temperature profiles is used to interpolate OHP and MOH lidar data from altitude to pressure level? Is it taken from reanalyses? ECMWF?*

The pressure and temperature data in the OHP lidar files are profiles of radiosondes launched in the nearby Nîmes station located at about 200 km, completed by NCEP operational data above the burst altitude of the balloon. At Hohenpeissenberg, temperature and pressure profiles from the co-located ozonesonde profiles (from MOH) or radiosonde profiles from Munich Oberschleissheim are used. Above $\sim 27$ km, the data is completed by co-located lidar temperature measurements, which are also used to extrapolate the radiosonde pressure data. We added a sentence to the manuscript to include this information.

[...] In case that the data were given in number density ($molecules\,cm^{-3}$), the VMR was calculated with the air pressure and temperature provided by the same instrument for ozonesondes or co-located ozone- or radiosonde data for lidars. For lidar measurements, these sonde temperature and pressure profiles are completed above the balloon burst by operational model data from the National Center for Environmental Prediction (NCEP) at OHP and by lidar temperature measurements and extrapolated radiosonde pressure data at MOH.

*p.7, lines 26-27. "...was corrected by its mean relative difference to GROMOS" . Was the mean calculated over the entire period of the GROMOS/GB instrument time series?*

Yes, the data were corrected by the mean value of the relative difference between GROMOS and the respective other instrument, by averaging the relative differences of the whole time period. This correction helped to better identify periods with anomalies, because constant offsets between GROMOS and another dataset were thus removed, and the same anomaly criteria could then be applied for all instruments. According to the comment of Referee #2 (p. 7, l. 25-27), this paragraph was adapted and an equation was inserted to clarify the matter.

To identify these anomalies we used a debiased relative difference ($RD_{debiased}$), given by

$$RD_{debiased,i,X} = RD_{i,X} - \overline{RD}_X, \tag{1}$$

where $\overline{RD}_X$ is the mean relative difference of GROMOS to the data set X over the whole period.

*p.7 line 29 "the respective month was identified as anomaly" Was anomaly identified in GROMOS data or in any of the compared instrumental records?*

The idea of the anomaly detection was to find out if GROMOS data show anomalies compared to the other instruments. We therefore only identify anomalies in the GROMOS records. The sentence was adapted to be more precise.

When this debiased relative difference was larger than 10 % for at least three instruments, the respective month was identified as anomaly in the GROMOS data.

*p.9 line 7 "the degrees of freedom of the trend fit" - can you please discuss this here - how you derive it.*

The degrees of freedom of the trend fit are the number of data points minus the number of fit variables, including relevant correlation terms. We adapted the text to specify this as followed.

The degrees of freedom are the number of data points minus the number of fitted variables. The latter are the number of the coefficients of the trend model plus the number of relevant correlation terms inferred by

the procedure described above.

> *p.9 line 21. "added to the corresponding part of covariance matrix" - can you provide more details? May be an example can be provided at this point xto demonstrate the process?*

A sentence with an example was added. A more detailed example with an example covariance matrix is given in von Clarmann et al. (2010).

For example, to account for a bias in the summer months of 2014, a fixed bias uncertainty is added to all variances and covariances of these months in $\mathbf{S_{bias}}$.

> *p.9 line 33-34. "value of the corresponding dataset at each altitude" - did you already remove bias earlier in the process? And thus, this will be the same value for all datasets, yes?*

For each trend profile, the mean profile of the corresponding instrument is used to determine the trend in percent per decade, so it is not the same value for all datasets. For example, to determine the trend in percent for the lidar at MOH, the model trend output for the MOH lidar (in ppm per decade) is divided by the mean ozone profile from the MOH lidar (in ppm). The sentence has been adapted as followed.

$$\text{Lidartrend}_{\text{MOH}}(\%) = \frac{\text{Lidartrend}_{\text{MOH}}(ppm)}{\text{Lidar}_{\text{MOH}}\text{ ozone mean value}} \cdot 100 \qquad (2)$$

The trends are always given in % per decade, which is determined at each altitude level from the regression model output in ppm per decade by dividing it for each data set by its ozone mean value of the whole period.

> *p.11, line 15-17 "this can be explained by the low number of OHP profiles..." - Why, if the two measurements sense the same ozone, there should be no large bias. Otherwise, what you are implying is that some large biases average out if one has a large number of coincident profiles. . .. I wonder if you should use coincident lidar measurements from OHP to evaluate if large biases between GROMOS and OHP sondes are indeed result of the spatial variability in ozone. Perhaps, you can plot PDF of subset of data (full or coincident) for both instruments to evaluate how they sample ozone variability over the same time-period. It will indicate if one instrument is not capable of obtaining low/high ozone, perhaps due to AK smoothing...*

The OHP ozonesonde seems to have some outlier profiles. For example, there seems to be a problem with the sonde temperature between Oct. 2002 and July 2007. The OHP ozonesonde data will be harmonized, but the reprocessed data are not available yet for the present study. We have now replaced a few profiles with a corrected version, but it does almost not affect the comparison. The ozonesondes and the lidar at OHP are usually not measuring at the same time. Therefore we can not compare coincident profiles directly. We adapted the text to clarify the influence of the small sample size of the OHP ozonesondes (see the comment p.11, l.15-17 of Referee # 2).

*p.11, line 30-33. You indicated that positive anomalies in the middle and lower stratosphere in 2000 were confirmed through anomaly detection method. You also found positive bias in the upper stratosphere in 2000, although it was not confirmed by comparisons with other instruments (MLS and SOMORA). Would the middle and lower stratosphere biases suggests the instrumental anomaly in upper stratosphere in GROMOS as it seems to affect the entire profile?*

You are right, an anomaly at a specific altitude might indeed affect also other altitudes in the GROMOS profiles. That's why we considered in the trend programme a bias at all altitudes as soon as an anomaly was detected at a specific altitude.

*p.12, line 21-22, why amplitude A is negative? Are coefficients in Eq (3) derived from observations? Why 0.1 ppm was selected for monthly uncertainty? Earlier you indicated that trend is 0.1 ppm per decade. What confidence level of the trend detection is expected in these time series?*

The amplitude A was now corrected to be positive. The coefficients $g$ and $h$ have been adapted to be negative, in order to obtain a realistic seasonal cycle with minimal ozone values in winter. The artificial time series was chosen in such a way that it is similar to a real ozone time series at around 10 hPa, where the mean GROMOS ozone value is approximately 7 ppm. The uncertainty of each monthly ozone value was set to 0.1 ppm, which was assumed to be a realistic uncertainty. This is is a bit smaller than GROMOS observational error due to thermal noise at 10 hPa. The confidence level for the resulting trend is 99%. If a larger uncertainty had been chosen, e.g. 0.4 ppm, the confidence level of the resulting trend would have been 95%.

*p.12, line 21-22 (continued), Also, please indicate the length of the artificial time series (yes, it can be deduced from Figure 5, but should be written in the first paragraph of section 4.1).*

We added the time period of the artificial time series in the text.

The artificial time series used for this purpose consists of monthly ozone values from Jan. 1997 to Dec. 2017 and is given by [...]

*p.12, line 29. "(∼0.4 ppm) " - should it be 0.35 or 0.36 based on a=7 ppm in Eq (3) and the rate of change in time series is very small (0.1/120 x 12 (month per year) x 20 (years) = 0.2ppm)*

Five percent of the average ozone value of 7 ppm would indeed be 0.35 ppm. However, the anomalies in the artificial time series were only added to summer months in the years 2014, 2015 and 2017. In the summer months the ozone values in the artificial time series for the corresponding years are approximately 7.9 ppm in June and 8.2 ppm in July and August. 5% thus correspond to 0.395 ppm in June and 0.41 ppm in July and August, or ∼0.4 ppm on average for each summer. We slightly changed the sentence to clarify that we only increase the ozone values for the summer months.

[...] We increased the monthly ozone values in the summer months (June, July, Aug.) of 2014, 2015 and 2017 by 5 % (≈ 0.4 ppm).

*p.13, line 4, you provide uncertainty now as 5 %? What is it in ppm? Is it still 0.35 ppm? Then 0.1 ppm uncertainty for series A was 1.4 %? It is a bit confusing to switch between % and ppm. Also, it is not clear what "corresponding months" means. Do you mean that uncertainties for JJA months in 2014, 2017 and 2017 were increased to 5 % while the rest of month in the entire time series has 0.1 ppm uncertainty?*

Yes, your description is correct, uncertainties were increased in the summer months of 2014, 2015 and 2017 to 5% of the mean ozone value (0.36ppm) and remained 0.1 ppm for the rest of the time series. We adapted the text and Table 2 to be clearer. The uncertainties are now always given in ppm, and only the added anomaly is given in percent. "Corresponding months" were replaced by "affected summer months".

[...] and set the uncertainties of the affected summer months to 0.36 ppm (case C in Fig. 6 and Table 2). This uncertainty value corresponds to 5 % of the overall mean ozone value. The uncertainty of the months without anomalies remained 0.1 ppm.

*p.13, line 12-13 you state that trend estimate improves " the higher the uncertainties are chosen". Did you test model with larger than 5 % uncertainties? Why 5 % was selected in the first place?*

Yes, we tested the trend model with uncertainties for anomalies larger than 5% and the bias uncertainties larger than 5 ppm. We decided to use 5% for the anomalies because it is a value close to the "real" added anomaly. The same was done later on for the GROMOS trend estimates, where the uncertainty for the anomalies was chosen based on the estimated "real" anomaly (compared to the other instruments), as described in Sect. 4.1.
Concerning the bias uncertainty (5 ppm), we made a sensitivity test and found out that 5 ppm is the smallest value where the trend remains constant. For larger bias uncertainties, the trend estimates remain the same. For smaller bias uncertainties, the a priori bias (which would be a bias of zero) gets more confidence which leads to different trend values. We therefore defined 5 ppm as the smallest bias uncertainty for which the bias is fully estimated from the data instead of being influenced by an a priori null bias.

*p.14 line 11-15, "GROMOS trend peak is observed at slightly lower altitude". " We observe a slight shift in the peak" of AK. - what do you mean? Can you explain the shift and when it occurred in time series? Or you mean that the maximum weight in AK profile for layer 35-40 km is higher by 2-4 km than the assumed peak of the AK at the middle of that layer?*

Since the new retrieval in 2009, the averaging kernel (AVK) peak does not always correspond exactly to the expected altitude. This means that at some pressure levels, the maximal amount of information is retrieved from slightly higher or lower altitudes than indicated. At pressure levels below ~10 hPa, the AVK peak is one or two pressure levels lower (~2-4km), above ~3 hPa the AVK peak is one or two pressure levels higher up (~2-4km) than expected (see Fig. 1 (c) in this response). We are aware of this problem and will try to solve it in a new retrieval version. For the present study, however, it is not problematic to use the data. The direct comparison with GROMOS is not affected by this problem because the data is convolved with GROMOS AVK. Only for the trend comparison, where no convolution is applied, it is important to mention this effect. We adapted the text to stress that it is a shift in altitude and not in time.

(Sect. 4.3:) We observe that after the instrument upgrade in 2009, the averaging kernels peak at higher altitudes than expected, [...].
(Sect. 4.4:) This difference might be related to the averaging kernels of GROMOS, which indicate that

[Figure]

Figure 1: Retrieval figures for a GROMOS retrieval from 02.02.2017. (a) Retrieved and a priori profile, (b) Averaging kernels (AVK) for specific pressure levels, (c) Peak of the AVKs, (d) Altitude resolution (full width at half maximum of the AVKs).

GROMOS retrieves information from higher altitudes than expected ($\sim 2\,km$ difference).

*p.14 line 11-15 (continued), Then, you also suggest that the full width of the AK is between 20 and 25 km. If it is centered at $\sim 36.5$ km, then you have to assume that the averaged (AK weighted) trend is estimated between 26.5 - 46.5 km altitude range.*

It's true that due to the altitude resolution, GROMOS trends at a given altitude are also affected by trends at other altitudes. However, this smoothing due to the averaging kernels (AVK) is not directly related to the altitude of the AVK peak. We therefore decided to remove the corresponding sentence ("Furthermore, one has to keep in mind the coarse altitude resolution of GROMOS, which is at 15 this altitude between 20 to 25 km.") and hope that it clarifies this paragraph.

[Figure]

[Figure]

(a) Example for a filter bench (FB) retrieval        (b) Example for a FFTS retrieval

Figure 2: GROMOS retrieved profile and corresponding AVKs for two sample dates before (a) and after (b) the instrument upgrade in 2009. In both situations, the atmospheric transmittance was 0.4.

> *p.14 line 11-15 (continued), Can you please provide AK example for the layer?*

Below we present an example for the GROMOS averaging kernels during the filter bench period (Fig. 2a in this response) and after the update to the FFTS (Fig. 2b in this response). The differences between the AVKs after the instrument upgrade have been analysed and are accounted for in the harmonisation (see comment below, p. 14, line 16-17).

> *p.14 line 11-15 (continued), If you apply the AK smoothing to the lidar profile and calculate trends. Would these trends be different from trends derived by integrating ozone profile between 35 and 40 km?*

Estimating the trends of the convolved lidar data and integrating it over the mentioned altitude range leads roughly to similar results. However, we did not consider trends from convolved data in the study, because trends are better represented when using the unconvolved data. The convolution might introduce artificial trends, especially towards the edges of the range where lidar data is available. Also, the above described change in the Gromos AVK after the instrument upgrade can introduce some artificial trends. This shift has been corrected in the GROMOS data by the harmonization, but it might introduce an artefact when estimating trends from convolved profiles from other instruments.

> *p.14, line 16-17. The harmonization of time series before/after 2009 (application of bias and seasonal cycle correction) does not necessarily means that the entire bias in the retrieved profiles will be removed. Say, the change in the instrument optical characterization would produce a shift in the vertical distribution of the AK weights, the upgraded instrument would start sampling at higher/lower altitudes. And since ozone trends can be different at higher/lower altitudes, it would affect the homogenized time series, even after you remove the step-change bias. Can you show the AK before and after 2009 (for approximately the same atmospheric conditions)?*

You are right that the instrument upgrade in 2009 led to changes in the AVK (see example cases in Fig. 2 in this response). These differences can also affect the time series and trends. To analyse this effect, we applied the mean AVK before and after the upgrade to a same ozone profile. The difference between the

[Figure]

Figure 3: Difference between the mean of filter bench (FB) profiles before 10-2009 and Fast Fourier Transform Spectrometer (FFTS) profiles after 10-2009 during the overlapping measurement period, used for the harmonization of FB data. The red line indicates the difference that is only caused by the difference in averaging kernels.

resulting convolved profiles shows the effect of the different AVKs on the profile (Fig. 3 in this response). We observed that this difference profile (bias due to AVK) looks similar to the difference profile that is applied to the data for the harmonization. The profile used for the harmonization has larger amplitudes in the peaks, indicating that additional differences are also corrected for. We conclude that the harmonization that we applied corrects for the effect that is due to differences in the AVK after the instrumental upgrade.

*p.15, line 1. Can the temporal sampling test be done on the artificial time series (change in sampling from 1hour to 3 minutes)?*

The temporal sampling should not affect the artificial time series as long as the diurnal and seasonal cycles are well represented.

*p.15, line 15-16. It is confusing to show "ozonesonde" results at altitude above 30 km. Maybe you should stop test at 30km.*

It might be indeed confusing that a trend profile based on ozonesonde measurements shows data above 30 km, even though only the ozonesonde measurement time and not the data itself is used. Keeping this information, however, gives information about possible effects in the upper stratosphere when an instrument measures only several times a week at noon. We therefore agree with Referee #2 (comment p.15, l.16) and propose to keep the figure as it is.

*p.16,line17-is it 45km or 42km?*

The trend peak of SOMORA is indeed at $\sim 1.8$ hPa ($\sim 43$ km), so we adapted the text.

We observe that all instruments that measure in the upper stratosphere show a trend maximum between $\sim 4$ and $\sim 1.8$ hPa (between $\sim 37$ km and 43 km), [...]

*p.16, line 22 "derive" - this sentence seems to have some missing text....*

Corrected, the word "derive" was misused and replaced by "deviate".

*p.16, line 30 - change "our" to "derived" or "estimated". Also "discrepant results" means "differences in derived trends"?*

The sentence was reformulated.

In the middle and lower stratosphere, at altitudes below 5 hPa ($\approx$ 36 km), the estimated trends differ from each other.

*Figures 6 b - add uncertainty bars to the results. You can also add circle slightly shifted horizontally to show where they fall.*

We adopted the figure as suggested.

*7 and 8 - please consider to change either green or blue colors - they are too close to be easily distinguished in the plot. Also, please consider to change uncertainty envelops from solid color to shaded or not even filled envelops - it is hard to distinguish between different instruments.*

We agree that the choice of color is difficult to discern, and we changed the colors to blue, red, and yellow. An important point in the trend profile figures is to discern if different trend profiles overlap within their uncertainties, and if they are significantly different from zero. It is less important, however, to discern the exact amplitude of the uncertainty. Therefore we decide to keep the uncertainty shading as it is, but added solid lines to the uncertainty envelopes, to be better distinguishable.

*10. - the same comment as above - selection of distinct colors and reduction in envelopes. May be the plot can be separate in two plots to reduce overlapping information - one for MOH and one for OHP.*

We adapted the colors and separated the figure in two subfigures as suggested.

**2  Author's response to Referee #2**

**2.1  General comment**

***Short resume:*** *Outlying measurements are a nuisance to many types of analysis. Time series analysis in particular can be very sensitive to outliers, impacting both the estimated regression coefficients and their uncertainty. This topic is especially relevant for ozone profile trend assessments as the actual ozone trend is expected to be quite small (not more than $\sim$ 2% per decade) which makes robust detection tricky and susceptible to a handful of outliers. This paper starts off by presenting a comprehensive analysis of periods of outlying measurements in the GROMOS time series. Where most other studies stop here and simply reject the outliers from further analysis, the authors apply a regression method proposed by von Clarmann et al (2010) which uses the timing of the detected/suspected outlying data and infers their magnitude along with other components in the regression model, such as the linear trend. Bernet et al. illustrate the value of the method using synthetic data, before applying it to the actual GROMOS time series. The methodology is explained well and the application is very relevant and adequately elaborated. The last part of the paper moves slightly away from the core subject of the paper, and briefly discusses the impact of sampling and length of the time series on the GROMOS trend result. These are also compared to trends obtained from neighbouring ground-based data records and Aura MLS, however, the adopted outlier detection and incorporation method are not for these data sets.*

***Recommendation:*** *This paper is very well written and the material presented supports the claims made by the authors. The subject is scientifically very relevant and fits the scope of ACP. I would therefore highly recommend to publish this paper as soon as my comments below have been incorporated.*

Thank you very much for your positive feedback and the recommendation. The detailed answers to your comments are given in the following

**2.2  Major comments**

**2.2.1  Order of vertical regridding and smoothing**

*p.7, l.7-9: The order of regridding and smoothing is unintuitive. First you downgrade highly resolved vertical profiles to a coarser grid, then convolve these with the MWR averaging kernel and a priori. In reality, the MWR instrument does not vertically smooth a coarse atmosphere. Shouldn't the procedure be the other way around? First smooth at best available vertical resolution, then regrid to the coarse grid. Hence you should oversample the averaging kernel in one direction to match the sampling resolution of the xh, as in Eqs. 10-11 of Keppens et al. (2015, AMT).*

We changed the AVK smoothing to the approach proposed by Keppens et al. (2015) (Eq. (15)) by interpolating the rows of the AVK to the highly resolved grid from the respective instrument. The changes compared to the approach used so far are small, but we recompiled all figures with this new approach and adapted the text in Sect. 3.1.2.

The profiles having higher vertical resolution than GROMOS were convolved by the averaging kernel matrix according to Connor et al. (1991), with

$$x_{conv} = x_a + \mathbf{AVK}(x_h - x_{a,h}) \tag{3}$$

where $x_{conv}$ is the resulting convolved profile, $x_a$ is the a priori profile used in the GROMOS retrieval, $x_{a,h}$ is the same a priori profile, but interpolated to the grid of the highly resolved measurement, $x_h$ is the profile of the highly resolved instrument, and **AVK** is the corresponding averaging kernel matrix from GROMOS. The rows of the **AVK** have been interpolated to the grid of the highly resolved instrument and scaled to conserve the measurement response (Keppens et al., 2015).

**2.2.2 Unconventional choice of definition relative difference**

*p.7, l.24: The relative difference is defined as $\Delta = 100(GRX)/GR$. Why did you not choose $\Delta' = 100(GRX)/X$, which is the more widely adopted definition? The ratio between both definitions is $\Delta/\Delta' = X/GR \geq 0$(you removed all negative values; p.6, l.24-25). Your expression will be more sensitive to detect $GR < X$, the conventional expression is more sensitive to detect $GR > X$. Did you test the sensitivity of the detected periods to this definition?*

We agree that the sensitivity for these two definitions is not the same. We therefore tested the anomaly detection also with the conventional definition and found that the two approaches lead only to small differences in the detected anomalies. We had chosen $\Delta = 100(GRX)/GR$ as definition for relative differences because the aim is to compare all the different instruments with GROMOS. We think that having GROMOS as a constant reference when calculating the relative differences is therefore more appropriate than having different references for each instrument comparison. Because of this advantage, and because the difference between both approaches is small, we decided to keep the definition as it is in the manuscript.

**2.2.3 Description far away from application**

*The description of the methodology (Sects. 2.2 and 2.3) is quite far away from the actual application. This makes it a slightly more difficult. You could consider explaining the outlier detection and regression methodologies right before their application in Sects. 3 and 4, rather than in Sect. 2.*

We followed your suggestion and changed the manuscript accordingly. We moved the comparison methodology in the beginning of Sect. 3 (Time series comparison), and the explanation of the trend model to Sect. 4 (Ozone trend estimations). Some section titles and the introductions of the affected sections have been adapted as followed.

2 Ozone data sets

The stratospheric ozone profile data used in the present study comes from different ground-based instruments that measure in Central Europe (Table 1). They are all part of the Network for the Detection of Atmospheric Composition Change (NDACC, 2018). In addition, we used data from the Microwave Limb Sounder (MLS) on board the Aura satellite (Aura/MLS). All data from the different stations are compared to data from the GROMOS radiometer located in Bern, Switzerland (46.95° N, 7.44° E, 574 m above sea level (a.s.l.)). The aerological station (MeteoSwiss) in Payerne (Switzerland, 46.8° N, 7.0° E, 491 m a.s.l.) is located 40 km south-west of Bern, which ensures comparable stratospheric measurements. The Meteorological Observatory Hohenpeissenberg (MOH, Germany, 47.8° N, 11.0° E, 980 m a.s.l.) is located 290 km north-east of Bern, and the Observatory of Haute Provence (OHP, France, 43.9° N, 5.7° E, 650 m a.s.l.) lies 360 km south-west of Bern. Even if stratospheric trace gases generally show small horizontal variability, the distance between the different stations, especially between MOH and OHP, may lead to some differences in measured ozone.

**2.2.4   Outliers are only detected & considered for GROMOS**

*You should state clearly that the proposed outlier detection and outlier incorporation methods are only applied to the analysis of the GROMOS time series. You could have done that for the other instruments as well, this would have resulted in a more fair comparison and improved the discussion in Sect. 4.5.*

We understand your remark that including the trend correction also for the other data sets would have improved the discussion. Unfortunately, applying the method to all data sets was beyond the scope of the current study. Further, GROMOS data show larger anomalies than most of the other dat sets (see e.g. their better agreement in Fig. 3), which makes the correction more urgent for GROMOS than for most of the other data sets. We added a sentence to clarify that the method was only applied to GROMOS data (see comment p.9, l.29).

**2.2.5   Show trend results when outliers are removed**

*Many, if not most, researchers remove outlying data points prior to regression. I am interested in seeing how this simple approach performs in the particular case of real GROMOS data (Sect. 4.2, Fig. 7) and of synthetic data (Sect. 4.1, Fig. 6). I acknowledge that this will not allow to draw conclusions that apply generally, since the result depends on the number and character of the outliers. At least this shows whether it was worth to invest extra coding time and analysis effort.*

We consider it a brute method to simply discard data of an anomalous data subset. To use the data in a conventional scheme or to discard them is sort of black and white view to the problem. Any additional additive component in the covariance matrix, regardless if only in the diagonal of $\mathbf{S}_y$ or a block encompassing multiple data points, will reduce the weight of the anomalous data, and thus we have the entire grey-scale between the "black" and "white" option available. This gives us greater flexibility.

Enhanced diagonal entries (case C in the artificial test time series) simply gives less weight to the anomalous data. Test cases D and E, via the correlated block in the covariance matrix, keep information on the relative differences between the anomalous data but largely ignore differences between the anomalous and the regular data. The latter approach performs better than simple downweighting. Since downweighting is a step towards discarding anomalous data, the poorer performance of setup C in comparison with D and E indicates that the covariance approach might indeed be better.

Rejection of anomalous data might be an option if the fraction of anomalous data is very small, but if a larger fraction of the data is subject to anomalies, we think that it is worthwhile to have a method which can save these data. Also, the trend uncertainties increase the more data is omitted in the time series. We concede that the method depends on two assumptions: (1) We need a priori knowledge on which data points have this anomaly, and (2) the anomaly must indeed be bias-like. This method (cases D or E) will not work for independent outliers, but we state these caveats in the paper.

Concerning the extra effort: The software tool has not been developed for this particular purpose. The motivation for the development was to be able to cope with inhomogeneous data sets biased against each other. But, having this tool available anyway, why not using it, since it has been shown to be useful also for the treatment of anomalies?

We also tested omitting anomalies in the artificial time series. This approach leads in this test data to similar results as the advanced uncertainty approach and is close to the true trend. This can be explained by the fact, that the artificial time series only consists of an ideal seasonal cycle without any noise, that is perfectly reproduced by the trend model, also if some data are missing. Nevertheless, even in this case omitting the anomalies leads to increased trend uncertainties. In a real time series, however, the data is less predictable. The GROMOS trend estimate when omitting anomalies is close to the trend estimate that simply weights the anomalies differently, but different from the corrected trend estimate where biases are fitted for anomalies.

We conclude that omitting data in the time series increases trend uncertainties and leads to a loss of information about data in the anomalous periods. We added the following paragraph to Section 4.1 to mention the disadvantage of omitting data.

A simple way to handle such anomalies would be to omit anomalous data in the time series. This, however, would increase trend uncertainties and lead to a loss of important information. Therefore, we use the presented approach to handle anomalous observations in the time series when estimating the trend. To account for these anomalies in the trend estimation, we make use of the fact that the user of the trend programme has several options to manipulate the error covariance matrix $\mathbf{S_y}$. [...]

**2.2.6   Expand discussion to trend uncertainty**

*The discussion in Sect. 4.1 is very insightful but does not treat another important factor in trend analysis: trend uncertainty. Can you elaborate by how much the outliers affect the trend uncertainty and whether the more advanced method is also able to provide more robust estimates of trend uncertainty? One element that determines trend uncertainty is the noise in the time series. The example shown in Sect. 4.1 is hence an idealised case without random uncertainties. The few added outliers will therefore have a larger impact on the trend estimate than in the more realistic case where noise would have been present. Have you*

*studied this effect? And please add a comment that the influence of outliers may be smaller in the case random errors would have been simulated as well.*

The trend uncertainty is estimated by the algorithm by propagating the $\mathbf{S_y}$ matrix onto the resulting trend. By default, noise in the data as well as autocorrelated error terms (which can be understood as atmospheric variability not covered by the trend model) are included. To account for anomalies in the data, we do not restrict ourselves to diagnosing their impact by a sensitivity study (case B) but we immunize the trend tool against anomalies, which is possible under the assumption that we know which data points are affected by the anomaly. For this purpose, we use the special feature of the trend program, which is that we can assign correlated uncertainties to subsets of the data. Just assigning a larger uncertainty to the anomaly-affected data points already reduces the impact of the anomaly from 33% down to 3%. Using the special feature to assign a fully correlated uncertainty (bias of unknown magnitude) to the data makes the trend estimate even more robust and the impact of the anomaly is almost fully removed. After this "immunization", the anomalies increase the trend uncertainty from $\pm$ 0.010399 ppm/dec to $\pm$ 0.010815 ppm/dec (last column in Table 2, but the precise digits are not shown in the table). These trend estimates include noise, the auto-correlated terms, and, for cases B to E, also the anomaly-induced uncertainties. This strategy might not have become clear from the original text. We have thus tried to rewrite parts of Section 4.1 (see Sect. 4.2 in the new manuscript). We do not agree that presence of noise in the data makes the effect of anomalies smaller. We can safely consider noise and anomalies as independent sources of error. Thus, their effects will add up quadratically.

**2.2.7  Positive trend at lower altitudes?**

*Please tone down the statement on p.14, l.33-35: "This result implies that ozone might also recover at lower altitudes, but the uncertainties, the dependency on anomalies, and the insignificance of the corrected trend show that this result is less robust." I read it as (1) there is a recovery (2) but the result is less robust. So, do you claim a recovery in this altitude region, or not? In fact, it is the region where GROMOS trends differ most from those obtained by other satellite- and ground-based techniques. Hence, more caution is needed in how this is phrased.*

We agree that a recovery can only be claimed if the result is robust. We therefore rephrased the sentence as followed.

The larger uncertainties in the lower stratosphere, the dependency on anomalies, and the insignificance of the corrected trend show that this positive trend in the lower stratosphere is less robust than the trend at higher altitudes.

**2.3 Minor comments**

*p.1, l.0: The title does not convey the main message of the paper. It should be improved so readers will find your work more easily. First, the manuscript is all about detecting unusual observations and taking these into account in the regression. Hence the "to consider uncertainties" is too vague, I believe you should be more specific. Uncertainties can mean anything, really. Second, the presence of "combining ground- based data" suggests that you merged data sets prior to regression, which is not the case. This is misleading, which is definitely undesired for a title of a paper. What about something along the lines of "Combined study of ground-based ozone profile data over Central Europe to detect and incorporate anomalous observations in the analysis of trends in the stratosphere"*

Thank you for your suggestions, the title has been adapted to better convey the main message.

(New title:) Ground-based ozone profiles over Central Europe: incorporate anomalous observations in the analysis of stratospheric ozone trends

*p.1, l.1-15: The abstract should be improved, as it is not clear exactly what the paper adds to the existing body of literature. For instance, "[...] an approach for handling suspicious anomalies [...]" does not say how you handle anomalies. This can be done in many ways. In my view, you should state clearly (1) that you analysed multiple ground-based data records to detect anomalous periods in the Bern MWR data record, and (2) that you then used this information to regress the Bern time series for long-term trends and the temporary bias for the detected anomalous data taking periods.*

We adapted the abstract to clarify the anomaly approach.

We present an approach for handling suspicious anomalies in trend estimations. For this, we analysed multiple ground-based stratospheric ozone records in Central Europe to identify anomalous periods in data from the Ground-based Millimetre-wave Ozone Spectrometer (GROMOS) located in Bern, Switzerland. The detected anomalies were then used to estimate ozone trends from the GROMOS time series by considering the anomalous observations in the regression. We compare our improved GROMOS trend estimate with results derived from the other ground-based ozone records (lidars, ozonesondes, and microwave radiometers), that are all part of the Network for the Detection of Atmospheric Composition Change (NDACC).

*p.1, l.18: "[...] in recent years [...]". Add LOTUS Report and replace (WMO, 2014) by (WMO, 2018) to the citation. Also, in the submitted version you sometimes refer to (WMO, 2014) and sometimes to (Pawson et al, 2014). Choose just one of these where relevant.*

We added the LOTUS report here, and changed Pawson et al. (2014) to WMO (2014) in the whole manuscript.

*p.2, l.6-8: Incorrect language. The linear trend fitted represents the linear component in the time series after all other natural variations are accounted for (solar, QBO, ENSO, aerosol, ...). This linear term is expected to receive contributions from decreasing ODS and increasing GHG, and possibly other unidentified processes. It is not possible to fit a linear term due to ODS and another linear term due to GHG, since these terms*

*are fully dependent. Hence, it is incorrect to state that "a trend might be masked by natural variability and factors such as GHG increases and changing dynamics". So this needs to be rephrased.*

The purpose of this sentence was not to state that a trend can be masked by factors such as GHG, but by natural variability. In addition we wanted to illustrate the complexity of ozone trend detection by mentioning some of the many processes that influence stratospheric ozone. We rephrased the sentence.

[...] detecting a small trend is a difficult task. Many factors influence stratospheric ozone such as variations in atmospheric dynamics, solar irradiance or volcanic aerosols and the increase of greenhouse gases (WMO, 2014). Further, ozone trends might be masked by natural variability. Other important sources for trend uncertainties are [...]

*p.2, l.10: Different in what sense? If you combine data records from different instruments, ok. But here such a merging is not considered. So a "changing resolution" or a "changing sampling pattern" is more relevant for a time series composed of measurements by one instrument.*

It's true that the first three aspects (instrumental drift, abrupt changes, biases) are reasons for trend uncertainties within a single time series, whereas the last point (sampling issues) is mostly relevant when comparing trends from different instruments. However, all of them can lead to differences between trend estimates from different data sets, which is the topic of this paragraph. We adapted the sentence as followed to clarify this aspect.

Other important sources for trend uncertainties are instrumental drifts, abrupt changes, biases, or sampling issues e.g. due to instrumental differences in sampling patterns or in vertical or temporal resolution.

*p.2, l.11: Add Frith et al. (2017) to the citation. The paper is more recent but also deals with ozone profiles which is closer to what you report in the manuscript.*

Corrected.

*p.2, l.12-14: The influence of outliers at start and end points on the trend estimates is well known and not discovered by Bai et al as suggested by the current phrasing. I propose to move the citation to the end of the phrase such as (e.g., Bai et al., 2017).*

Corrected as suggested.

Biases in ozone data sets can lead to important differences in trend estimates, especially when they are occurring in the beginning or the end of the considered trend period (e.g. Bai et al., 2017).

*p.2, l.20: I am not sure there is evidence that all data sets used by Steinbrecht et al. (2017) have been successfully corrected for drift. Hubert et al. reported drift in parts of the HALOE, SCIAMACHY and GOMOS data sets. These have not been corrected. And so far no one has proven that the OSIRIS correction is removing the entire drift. Please rephrase this sentence.*

Compared to older studies (Harris et al., 2015; WMO, 2014), Steinbrecht et al. used newer data sets with reduced drifts (e.g. SAGE-OSIRIS) and omitted older data sets with large drifts (SAGE-GOMOS), see

Table 3 in Steinbrecht et al. (2017). We rephrased the sentence indicating that they used "data sets with small drifts", instead of "drift corrected data sets".

The study of Steinbrecht et al. (2017) summarizes recent trend estimates using only updated satellite data sets with small drifts.

*p.2, l.34: Add a reference to the LOTUS report as it contains the most recent and complete set of trend estimates.*

We added the LOTUS report as reference.

*p.3, l.1: Confusing language "handle [...] anomalies by considering anomalies [... ]". It would help to define up front what you mean with "anomaly". Is it due to the instrument, due to natural processes, ...? A great deal of readers will read anomaly as something that is not problematic, as it is caused by natural processes. Others will interpret it as an issue to solve.*

We adapted the sentence and defined "anomalies" in the introduction (see comment below, p.3, l.12).

The present study proposes an approach to handle the problem of anomalous observations in time series by considering the anomalies when estimating trends.

*p.3, l.5: "[...] resulting in a corrected trend estimate [...]". Corrected with respect to what? Perhaps "improved" is more in place here?*

We agree that in this part of the manuscript, an "improved trend" is more appropriate to stress the fact of the improvement. We changed the term from "corrected trend" to "improved trend" in this paragraph. Later on, however, the correction of the anomalies in the trend estimation is described (e.g. Sect. 4.2), which illustrates that the trends are corrected with respect to the trends where anomalies were not considered. We therefore decided to keep the term "corrected trend" in the part of the manuscript that follows the description of how this "correction" is performed.

*p.3, l.12: The word "anomaly" is often also used for anomalous geophysical events. This class of anomalies is not the subject of this manuscript. In caption of Fig. 1 you use "anomaly" in the sense of a "deviation from a climatology", which agrees with how many readers will interpret it. To avoid any confusion I suggest to define "anomaly" unambiguously at the beginning of the manuscript (and perhaps repeated here and there in the text). And are inhomogeneity (see comment below) and anomaly the same for you?*

We now defined anomalies in the introduction and repeated it later on. We also removed the word "anomaly" in the caption of Fig. 1 and in the corresponding text (Sect. 3.2) to avoid confusion. Concerning the inhomogeneities, see comment below (p. 8, l.15).

*(Introduction:) We define anomalies as periods where the data deviates from the other data sets.*
*(Sect. 3.2:) This ozone deviation is calculated by the ratio of the deseasonalised monthly means (difference between each individual monthly mean and the corresponding climatology of this month) and the monthly climatology.*

*p.3, l.11: Section 2 is a tad long. If you consider trimming the manuscript (which is not what I propose), then this section would be a good place.*

We agree that the data and methods section is a bit long, but we think that it is important to give information about the used instruments. Also, it is essential for the study to explain details of the trend programme and how it is used to account for anomalies in the trend estimates. However, following your suggestion, we reordered the method part by moving the time series and trend description to the respective section (see major comment 3).

*p.4, l.16: Add "mean" or "median" to the moving window to clarify what statistical measure you used for the location of the smoothed data.*

"Moving window" has been replaced by "moving median window" (see also comment from Referee #1, p.4, line 16 (or line 18)).

*p.5, l.17: Replace by "[...] using concurrent total column ozone data from the Dobson [...]". This to clarify that the corrections are done using data taken at the same/similar time.*

Corrected.

The profiles are normalised using concurrent total column ozone from the Dobson spectrometer at Arosa [...]

*p.6, l.4: You mentioned the spectral line for the ground-based MWR, so it would be nice to have the frequency of the line for Aura MLS as well.*

We agree and adapted the manuscript.

It provides profiles of different trace gases in the atmosphere, with a vertical resolution of $\sim 3\,\text{km}$. Stratospheric ozone is retrieved by using the spectral band centred at 240 GHz.

*p.6, l.7: Improve language "stable with insignificant drifts", this is twice the same information. Either use "stable", or more precisely "stable within X% per decade", or just "because there are no drifts between".*

Corrected.

We chose the Aura/MLS data for our study because there are no drifts between 20 and 40 km (Hubert et al., 2016).

*p.6, l.9: Clarify what data is compared. E.g. "Comparison of time series". (Further on in the paper there is also a comparison of the trends.)*

After moving this subsection to Sect. 3 (Time series comparison) according to your suggestion in major comment 3, we changed the title of this subsection to "Comparison methodology". Because this is a subsection of the section "Time series comparison", it is not necessary to include "time series" again.

*p.6, l.17: Why is the GROMOS retrieval accurate between 31-0.8 hPa? The retrieval is not accurate simply because the measurement response is high. The instrument can still be biased (large systematic error) and even imprecise (large random uncertainty). The use of "accurate" suggests a small uncertainty. Is that really the case?*

"Accurate" is indeed not the appropriate wording. We adapted the sentence.

In this study we concentrate on the altitude range between 1 and 0.8 hPa, where the a priori contribution to GROMOS profiles is low (see Sect. 2.1).

*p.6, l.22-24: The native measurement coordinate system for lidar is number density versus altitude. You need to explain how the lidar data are converted as well. For ozonesonde this is trivial due to the temperature data from the attached radiosonde.*

See the comment of Referee #1, p.6, l. 27.

*p.6, l.26-29: Please clarify the details of the regridding procedure. The description is confusing and I am not sure I understand the order of operations and on what observed quantities.*

We reformulated the sentence and hope that it is clearer now. To have similar grids for the different instruments, we interpolated the altitude-based data first to a regular altitude grid, and averaged then the provided pressure data to obtain a pressure grid with the same grid points.

The lidar and ozonesonde data were therefore linearly interpolated to a regular spaced altitude grid of 100 m for the ozonesonde at OHP and Payerne, and 300 m for the lidars and the ozonesonde at MOH. The mean profile of the interpolated pressure data then built the new pressure grid for the ozone data.

*p.6, l.26-29: Also, is the interpolation a simple linear interpolation or is it a more advanced method such as described in Calisesi et al. (2005)?*

We used a simple linear interpolation and added this in the manuscript.

The lidar and ozonesonde data were therefore linearly interpolated to [...]

*p.7, l.3 and p.7, l.21: Remove "direct" from "direct comparison", since there are no indirect comparisons in this manuscript.*

We changed the subsection title to "GROMOS comparison and anomalies".

*p.7, l.25-27: The use of "corrected" relative difference is misleading. Why not e.g. "debiased" relative difference or "variations of the" relative difference? You simply remove the mean level to look at deviations of the relative difference w.r.t. its mean level. Please abstain from using "corrected relative difference" elsewhere in the text as well.*

We changed it to "debiased relative difference" and added a formula to the text.

Based on the relative differences we identified periods where GROMOS differs from the other instruments. To identify these anomalies we used a debiased relative difference ($RD_{debiased}$), given by

$$RD_{debiased,i,X} = RD_{i,X} - \overline{RD}_X,\qquad(4)$$

where $\overline{RD}_X$ is the mean relative difference of GROMOS to the data set X over the whole period.

*p.7, l.25-27: Add formula to clarify that you remove a constant offset in RD. In other words, assuming RDi is the monthly mean of month i and the mean of RD over all months is μRD, then you debiased the RD time series using RDi - μRD.*

We added a formula and reformulated the text.

For comparison with GROMOS we computed relative differences between the monthly mean values of the different data sets and the monthly mean of the coincident GROMOS profiles. The relative difference (RD) for a specific month $i$ has been calculated by subtracting the monthly ozone value of the data set ($X_i$) from the corresponding GROMOS monthly mean ($GR_i$), using the GROMOS monthly mean as a reference:

$$RD_{i,X} = (GR_i - X_i)/GR_i \cdot 100.\qquad(5)$$

*p.8, l.19-20: Von Clarmann present an approach to consider "known", "identified" or "suspected" inhomogeneities in the trend analysis. Their method does not identify the time of an inhomogeneity, but co-fits the magnitude of the temporal offset. Second remark, state right here that you apply the von Clarmann method and code.*

We added "known" and "suspected" to the sentence. We added a sentence to clarify that we use the method from von Clarmann et al. (2010) in our analyses.

Von Clarmann et al. present an approach to consider known or suspected inhomogeneities in the trend analysis. [...] We use the method and code from von Clarmann et al. in our trend analyses [...].

*p.8, l.15-...: Now you start to use "inhomogeneities" rather than "anomalies". Is there a difference between these terms? Please define these clearly, if they are. If not, then state that as well.*

"Inhomogeneities" is the general term that we use for the method by von Clarmann et al. (2010). As you mentioned in your comment below (p.8, l.21-22), this can include known causes such as changes in measurement systems or sampling irregularities, but also unknown instrumental issues. The unknown instrumental issues in our data lead to inhomogeneities in some months. We call these inhomogeneities in some months *anomalies*. We added the following sentence to clarify this.

We use the method and code from von Clarmann et al. in our trend analyses to account for inhomogeneities. The inhomogeneities are in our case anomalies in some months that we identified as described in Sect. 3.1.2.

> *p.8., l.15: Strengthen the link with previous Section 2.2. State asap that you will now use the times of the anomalies detected by the previously described method (Sect. 2.2) in the regression scheme (Sect. 2.3).*

We adapted the manuscript and reformulated parts of the paragraph.

The inhomogeneities are in our case anomalies in some months that we identified as described in Sect. 3.1.2. [...]
To account for the anomalies in the time series when estimating the trends we adapted $S_y$ in two steps. First, we increased the uncertainties for months and altitudes where anomalies were identified (using the method described in Sect. 3.1.2).

> *p.8, l.15-16 and/or l.23-24: Clarify that the full error covariance matrix describes the correlation between measurements, not between pressure levels.*

Corrected.

The total uncertainty of the data set is represented by a full error covariance matrix $S_y$ that describes for each pressure level covariances between the measurements in time.

> *p.8, l.21-22: Remove the entire phrase "Another reason for [...]". Instead add this possible cause (unknown origin) to the end of the phrase on l.16-18 "Inhomogeneous data can originate from [...] spatial or temporal sampling." (which currently lists known causes). The method you described in Sect 2.2 identifies periods of anomalies, but does not identify the cause of the anomaly. It can be anything in the list of l.16-18, but also unknown instrumental issues.*

Corrected.

Inhomogeneous data can originate from changes in the measurement system (e.g. changes in calibration standards or merging of data sets with different instrumental modes), from irregularities in spatial or temporal sampling, or from unknown instrumental issues.

> *p.9, l.5: "no error correlation", but in what domain? Clarify that you deal with the correlation between measurements, not between pressure levels.*

Corrected.

The off-diagonal elements of $S_{instr}$ are set to zero, assuming no error correlation between the measurements in time.

> *p.9, l.5: You introduce a second VCM (variance-covariance matrix). It would help to have a separate symbol for the different VCM's (S1, S2, Sinstr,Sautocorr or whatever) and use in-line formulas to explain when you add these and when not. This allows you to get away from unclear language such as "additional covariance matrix" and "enhanced covariance matrix". You could just refer to it as S' = S1 + S2. This also*

We introduce as suggested different symbols for the different covariance matrices and include the corresponding equation in the manuscript.

The covariance matrix is for each instrument given by

$$S_y = S_{instr} + S_{autocorr} + S_{bias} \qquad (6)$$

where $\mathbf{S_{instr}}$ gives the monthly uncertainty estimates for the instrument, $\mathbf{S_{autocorr}}$ accounts for residuals autocorrelated in time which are caused by atmospheric variation not captured by the trend model, and $\mathbf{S_{bias}}$ describes the bias uncertainties when a bias is considered.

*p.9, l.12: Clarify "which means the time correlated residuals" are not considered (S2 or Sautocorr ).*

[...] which means that the time correlated residuals ($\mathbf{S_{autocorr}}$) are not considered for the GROMOS trend.

*p.9, l.15: Now you switch back to using "anomalies", not "inhomogeneities". Be consistent or mention that both terms have identical meaning.*

See comment p.8, l.15-...

*p.9, l.16: It is important to link with previous section. "[...] for months where anomalies were identified (using the method described in Sect. 2.2)".*

First, we increased the uncertainties for months where anomalies were identified (using the method described in Sect. 3.1.2.

*p.9, l.18: Are you using the so-called "corrected relative differences" or not?*

Yes, we used the mean of all debiased relative differences, with the aim to obtain a representative value for each anomaly. We adapted the text to be clear.

For this purpose, we set the diagonal elements of $\mathbf{S_{instr}}$ for the respective month $i$ to a value obtained from the mean difference to all instruments ($RD_{debiased,i}$) and the mean GROMOS ozone value.

*p.9, l.24: Replace "low" values by "small" values. E.g. -25% is a very low value, but not a small value when compared to -5%. I know the example is not applicable here, but I would generally suggest to refrain from using "low/high" and use "small/large" instead when referring to the magnitude of values (not the sign).*

Corrected.

*p.9, l.29: Clearly state that above procedure is only applied to GROMOS data. You did not consider outliers in any other time series than GROMOS.*

We added the following sentence to the beginning of the paragraph. Further, we clarified this again in the beginning of the section about trend comparison (Sect. 4.4).

(Sect. 4.1:) The described procedure for anomaly consideration is only applied to the GROMOS time

series, whereas the other trend estimates were not corrected for anomalies.

**4.4 Trend comparison**

The GROMOS trend profile (corrected as described above) and trend profiles for all instruments at the other measurement stations (uncorrected) are shown in Fig. 10.

*p.9, l.32-34: The percentage trend is obtained by dividing the ppmv trend by the mean level. If there is a trend the mean value will depend on the period over which the mean is computed. Did you use exactly same period for all trends? Since Aura/MLS has the shortest record, I would guess the ppmv trend should be scaled to 2005-2017 mean values. Has this been done? If not, please state whether this effect is negligible.*

We did indeed not use the same time period to calculate the percentage trend for the different data sets. However, we tested the difference and found that the effect is negligible. For GROMOS, for example, the difference between the percentage trend calculated over the whole period and over the period 2005-2017 (period of Aura/MLS trends) is smaller than 0.05 throughout the whole profile, which corresponds to a difference between both trend profiles of ∼1 % in the middle stratosphere and 2 % at the edge of the upper stratosphere. This is a very small differences for a trend of ∼2%, and we therefore decide to use the available time series period for each data set, as it was done so far.

*p.10, l.1: You could remove ", as described by Tiao et al. (1990)", as this is just introductory-level Gaussian statistics.*

Corrected.

*p.11, l.1-2: You refer to "another temporal sampling", another than the other instruments? Another than the usual GROMOS sampling?*

Shown in Fig. 2 is the complete GROMOS time series with its full sampling (30min to 1h), and not only coincident data with one of the other instruments. We adapted the sentence.

[...] whereas the GROMOS data shown here represent the complete GROMOS time series with its high temporal sampling.

*p.11, l.9-10: Remove ambiguity. You deal with relative differences of monthly mean values of time coincident pairs, not of the monthly mean value of the relative difference of time coincident pairs. Figure 3 suggests the latter however.*

We adapted the figure captions (Fig. 3 and Fig. 4) to "Relative differences of monthly means" and refer to the new Eq. (2) in the text.

Shown are the monthly relative differences of time coincident pairs of GROMOS (GR) and the convolved data set X, with GROMOS data as reference value (Eq. (2)).

*p.11, l.15-17: The large outliers in the OHP comparison time series (Fig. 3) are all negative. The explanation is not satisfying, if not incorrect. If outliers are due to the small sample, then large positive peaks should be observed as well. Unless there is a systematic component in both the sampling pattern and the ozone fields. The latter seems highly unlikely to me. Can't it, instead, be due to the presence of large positive outliers in the OHP ozonesonde record?*

The small sample is indeed not the cause for the differences, but it makes them more visible when comparing monthly means. Having for example one ozonesonde profile with large differences to Gromos within a month and several profiles with very good agreement, the monthly mean would not be affected much by the one profile with bad agreement. However, when we have in some cases only one or two profiles per month, the monthly mean is highly affected in case that the difference to GROMOS is large for these profiles. We reformulated the text to clarify this aspect. See also the comment of Referee #1, p. 11, l. 15-17.

We further observe that the relative difference between GROMOS and the OHP ozonesonde shows some important peaks in the last decade, indicating that the ozonesonde often measures more ozone than GROMOS. The ozonesonde data seems to have some outlier profiles. When comparing the monthly means of coincident pairs, these outliers are even more visible, because only a small number of OHP ozonesonde profiles are available per month (only four profiles on average).

*p.11, l.9: Does Fig. 3 show the "corrected relative difference"? I guess not, since the mean level seems to differ from zero. This should be clarified.*

Fig. 3 shows indeed the "normal" relative difference (RD). The debiased relative difference (RD$_{debiased}$) was only used to detect the anomalies, but is not shown in the figures. This makes it possible to see the "true" differences between the instruments in the figures. The debiased relative difference is just a correction of RD to make the anomaly detection easier, by removing a potential constant offset. This has been clarified by using the acronyms RD and RD$_{debiased}$ in the figure captions of Fig. 3 and Fig. 4.

*p.11, l.21: Doesn't Fig. 4 show the "corrected relative difference" (or debiased relative difference, see earlier comment above)? Mention this clearly in the text, and abstain from using "corrected relative difference", I like "debiased relative difference" better.*

See comment above. To clarify we modified the caption of Fig. 4.

The black areas in the lowest panel show periods where at least three data sets (or two data sets above 2 hPa) have a debiased relative difference (RD$_{debiased}$) larger than 10 %.

*p.11, l.34: Replace by "[...] comparing GROMOS to SOMORA [....]". There is only one reference instrument in the US during the period before Aura/MLS.*

Corrected.

[...] were not confirmed by comparing GROMOS to SOMORA, and might thus be real ozone variations.

*p.12, l.1: Add "[...] at these altitudes, especially prior to the start of the MLS measurements in 2004." And you could then e.g. also mention the step change around 2005 visible in panel (a) around 2005 which may be related to the change of SOMORA front end (see also comment below; p.28, Fig.3).*

We added the proposed sentence. Concerning the step change, see comment below (p. 28, Fig. 3).

[...] which makes the anomaly detection less robust at these altitudes, especially prior to the start of Aura/MLS measurements in 2004.

*p.12, l-7.8: It is not entirely clear whether you are saying here that you disprove the claim by Steinbrecht (2009)? If yes, then end the phrase by removing the ambiguity. For instance: "[...] due to instrumental issues of GROMOS and not due to sampling."*

We adapted the text as proposed.

[...] which are probably due to instrumental issues of GROMOS and not only due to sampling differences.

*p.12, l.7-8: Are there any leads on what could be the cause of the instrumental issues? If yes, it may be of interest to the reader.*

We checked several parameters such as instrumental problems, noise temperature, or atmospheric opacity but could not find any clear explanation for the anomalies so far.

*p.12, l.21: A negative amplitude has no meaning. Is this a typo?*

The amplitude should indeed be positive, it was adapted so that $g$ and $h$ are negative, in order to obtain the desired phase of the seasonal cycle (see also comment of Referee #1, p.12, l. 21-22).

*p.12, l.26: Vague language "This might be due to some error propagation". This clause does not hold additional information and may as well be removed.*

We removed the sentence as proposed.

*p.13, l.4: You arbitrarily chose 5%, which is 3.5 = 5/(0.1/7) times larger than 1.4% = 0.1/7. What if you increase to even higher uncertainties, e.g. in the limit of infinite uncertainty? This case would be equivalent to removing the anomaly from the time series prior to regression. (See also my major comment)*

Indeed, increasing the uncertainty would improve the trend estimate (as mentioned in the manuscript), especially in the perfect, noise-free artificial time series. However, we decided to use a realistic value for this uncertainty, since the trend is only affected marginally when using an even higher uncertainty (see your comment below, p.13, l.34). See also the comment of Referee #1 (p.13, line 12-13) concerning the choice of the uncertainty and your major comment concerning the option to remove the anomalies.

*p.13, l.6: Text is inconsistent with caption of Fig. 6. Shouldn't it be "[...] once without (case D) and once with (case E) the increased [...]"?*

You are right, we corrected the text.

*p.13, l.7: Add to the last phrase that a 5 ppm correlated block essentially leads to a free fit of the magnitude bias.*

Adding the correlated block to $\mathbf{S_y}$ corresponds to an unknown bias of the data subset that is affected by anomalies, and leads to a free fit of the bias magnitude. This bias is represented in $\mathbf{S_y}$ as a fully correlated block of $(5\,\text{ppm})^2$. It has an expectation value of zero and an uncertainty of $5\,\text{ppm}$.

*p.13, l.27: Repeat by how much you increased the uncertainty, so a reader does not have to go back all the way to Sect 2.3.*

[...] we increased the uncertainties (diagonal elements of $\mathbf{S_y}$) for the months and altitudes that were detected as anomalies. The uncertainties for these anomalous data have been set to a value obtained from the difference to the other datasets ($\text{RD}_{debiased}$), as described in Sect. 4.1.

*p.13, l.34: "[...] decreases the trend slightly in the lower stratosphere, but the differences are small." Doesn't this point to a too small scaling of the uncertainties?*

The uncertainties for the anomalies has been set to a realistic value based on the relative difference detected by comparing GROMOS data to data from the other instruments. Setting them to higher values affects the trend only slightly. Therefore we propose the more advanced approach, where a bias is fitted by considering a correlated block in the covariance matrix.

*p.14, l.4: I still don't like the phrasing "corrected" trend profile. Corrected w.r.t. what? See also my comment p.3, l.5.*

See the answer to your comment for p.3, l.5.

*p.14, l.4 and l.30-31: "The corrected trend profile agrees well with recent satellite-based ozone trends [...]". I am not convinced that the case III results agree well with other studies. They are consistent with Steinbrecht (2017; Fig.5, Table 6) between 10-2 hPa (what you call "middle" stratosphere). However, there is tension between the results at 30-10 hPa ("lower" stratosphere). The other studies find smaller trend values, more closely to 0-1% per decade.*

Adapted as followed.

[...] The corrected trend profile is consistent with recent satellite-based ozone trends (e.g. Steinbrecht et al., 2017) in the middle stratosphere.

Compared to other studies, the GROMOS trends in the lower stratosphere are slightly larger than trends of most merged satellite data sets.

[...] This is consistent with satellite-based trends from recent studies in the middle stratosphere, but is still larger than most satellite trends in the lower stratosphere.

*p.14, l.13-14: Somewhat confusing phrasing "a slight shift in the peak height". A shift with respect to what? Is the peak shifting in time, which is what I first think of in the context of time series analysis? More likely you mean that the peak sensitivity of the GROMOS retrievals is not at the true altitude?*

Text adapted, see also comment of Referee #1 (p.14 line 11-15).

(Sect. 4.3:) We observe that after the instrument upgrade in 2009, the averaging kernels peak at higher altitudes than expected, [...].

(Sect. 4.4:) This difference might be related to the averaging kernels of GROMOS, which indicate that GROMOS retrieves information from higher altitudes than expected ($\sim 2$ km difference).

*p.14, l.30-31: See earlier comment, I would refrain from using "agree". The results are consistent, but do not agree well especially below the 10 hPa level. The bias corrected trend does lead to a better agreement*

In summary, correcting the GROMOS trend with our anomaly approach leads to a trend profile of $\sim 2$ to 2.5 % per decade in the lower and middle stratosphere. This is consistent with satellite-based trends from recent studies in the middle stratosphere, but is still larger than most satellite trends in the lower stratosphere.

*p.15, l.16: Unlike Reviewer 1, I like the fact that the full altitude range is shown. This section is not about a comparison to actual ozonesonde or lidar trends, but about the impact of sampling on GROMOS trends. I would keep the altitude range in Fig. 8 as is.*

Thank you for this statement, we agree and keep it as it is (and expanded it to other instruments, see next comment).

*p.15, l.17-19: This is a very nice piece of information. Out of curiosity, have you tested to randomly subsample the GROMOS time series to MOH and NPay, then regress? Such an analysis would add information about the random nature of the impact of sampling. The more positive sonde trend at 3 hPa may as well turn into a less positive trend if another temporal GROMOS sampling was picked but with same sample size as the ozonesonde time series. Would be nice, but not necessary.*

Thank you for this input. To check how the trends look like with other sampling rates, we added also trends with sampling rates of the other lidar (OHP) and ozonesondes (MOH and OHP). The larger trends at 40 km when using only data at ozonesonde times is confirmed by the additional sampling rates of the other two ozonesondes. We adapted the figure in the manuscript and the text as followed.

(Sect. 4.3.1:) The GROMOS time series is used for these trend estimates, once using only measurements at the time of lidar measurements, and once only at the time of ozonesonde launches. The differences to the trend that uses the complete GROMOS sampling are not significant, but still important, especially between 35 and 40 km and in the lower stratosphere. Using the sampling of the MOH lidar leads to larger trends ($\sim 3$ % per decade) than using the normal sampling ($\sim 2$ % per decade) below 5 hPa. The OHP lidar

sampling, however, leads to smaller trends than the normal sampling above 10 hPa. This suggests that selecting different night measurements within a month can lead to trend differences.

All three ozonesonde samplings result in a larger trend than normal or lidar sampling above 5 hPa. [...]

(Sect. 4.4:) [...] distance of $\sim 600$ km between the MOH and OHP stations might also explain some differences between the lidar trends. Furthermore, our sampling results show that the lidar sampling at MOH leads to a larger trend in the lower stratosphere than using a continuous sampling, whereas OHP lidar sampling leads to a lower trend in the middle stratosphere. The large lidar trend at MOH and the comparable low OHP lidar trend might therefore also be partly explained by the sampling rate of the lidars.

*p.15, l.26: Replace "estimations" by "regression".*

Corrected.

*p.15, l.26-27: Did you average the GROMOS trends or the GROMOS time series? If the former, how did you do that and does the trend of vertically averaged data agree with the vertical average of the trends?*

We averaged the GROMOS trends over the three different pressure ranges. We think that this should be quasi the same as the trend of vertically averaged data.

*p.16, l.5-6: Vague language "The trend dependency [...] is controversial to [...]". Please rephrase, I do not understand what you mean.*

The trend magnitudes depend on the starting year of the regression, which is controversial to the definition of a linear trend that does not change with time.

*p.16, l.6-7: "This suggests that the true trend might not be linear, or that some interannual variations or anomalies are not captured by the trend model" This is known for a long time. Please rephrase to "This illustrates that [...]".*

Corrected.

*p.17, l.2-3: "A thorough harmonization would be necessary to correct the trend for this change." In fact, your manuscript describes a method that does not require harmonisation. I understand the extra work effort, but it is a pity for Section 4.5 that you did not treat outlying data in the non-GROMOS regression analyses. This would have made an even more interesting comparison.*

We agree that it would be a good idea to improve the trends also for the other data sets. Nevertheless, it was more urgent to consider anomalies in the GROMOS data because there are some clear anomalies compared to the other data. See also major comment 4.

*p.17, l.4: It is unclear whether you have used these newer SOMORA retrievals or not. Please clarify in the revised manuscript.*

The SOMORA trend shows a positive peak at 30 km, which is probably due to homogenization problems that are corrected in the new retrieval version of SOMORA, which is, however, not yet used in our analyses (Maillard Barras et al., 2019).

*p.26, Fig.1: "[...] indicate months without data." Missing data does not imply there are no data, and you mean there are no (screened) data during these months.*

The white lines indicate months where no measurements were available due to instrumental issues.

*p.26, Fig.1: Mention the period over which the climatology is computed. So, "Deviation from GROMOS monthly mean 1997-2017 climatology" would make a better description.*

Corrected.

Deviation from GROMOS monthly mean climatology (1997 to 2017), smoothed by a moving median window of 3 months.

*p.28, Fig.3: "Monthly means of relative differences" implies you compute relative difference 100 x (GR-X)/GR for each coincident pair, then take the monthly average. This is not how you described it in Sect. 2.2.*

We adapted this wrong formulation, see comment above (comment p.11, l.9-10).

*p.28, Fig.3: Do these time series represent the "relative difference" or the "corrected relative difference"? If these are corrected relative differences, then how can it be that most curves seem to have a negative mean value where zero would be expected?*

They represent the "uncorrected" relative difference, see comment above (comment p.11, l.9).

*p.28, Fig.3: Panel (a) shows a step around 2004-2006. Is this due to the change of front end in SOMORA in 2005 (p.4, l.21)? If yes, then it would be nice to at least mention this in the last paragraph of p.11.*

This step change might indeed be related to the front-end change of SOMORA in 2005. We added the following sentence to the manuscript.

A step change between SOMORA and GROMOS is visible in 2005, which might be related to the SOMORA front-end change in 2005.

*p.29, Fig.4: Do all panels represent the "relative difference" or the "corrected relative difference"? Clarify in the caption in the latter case.*

All panels show the relative differences – and not the debiased (corrected) relative differences – to show the true difference to GROMOS. The debiased relative difference has only been used to detect the anomalies (see also the answer to comment p.11, l.9).

*p.34, Fig.9: See my earlier comment. Did you average the trend, or average the time series then derive the trend? In the first case, how exactly did you compute the error on averaged trend?*

We averaged the trends and the uncertainties after estimating them from the time series (see also comment p.15, l.26-27). Since the different trend profile points that are averaged for a specific pressure range are not independent, we decided to use the arithmetic mean of the uncertainties, which leads to slightly larger uncertainties than using a standard error (standard deviation divided by the square root of the sample size).

*p.34, Fig.9: Mention significance level of the error bars. I assume that these represent $1\sigma$ since some bars that do not cross the zero level are greyed out.*

We adapted the figure to show the $2\sigma$ uncertainty and mentioned it in the figure caption.

*p.37, Table.2: Add time unit in first column. It should be "ppm/decade".*

Corrected.

*p.37, Table.2: Move third column ("Monthly uncert.") to first column in section "Parameters in the trend programme". No uncertainty has been added in the time series.*

Corrected.

*p.37, Table.2: Use consistent number of significant digits. I suggest three digits for all numbers quoted.*

We agree and adapted the number of digits to be consistent for the estimated trends in Table 2. For the other numbers, we kept it as it is because we do not think that it is necessary to add zeros to all of them.

**2.4 Technical corrections**

We adapted the manuscript to the suggested technical corrections and show the changes in the following.

> *p.1, l.4-5: Replace by "[...] anomalies that may all mask a [...]".*

Trends and their uncertainties are influenced by factors such as instrumental drifts, sampling patterns, discontinuities, biases, or short-term anomalies that may all mask a potential ozone recovery.

> *p.1, l.7: Remove "[...] to improve the derived trend profiles [...]". This is evident.*

We present an approach for handling suspicious anomalies in trend estimations.

> *p.1, l.11: Replace by "[...] in agreement with satellite [...]".*

[...] providing a confirmation of ozone recovery in the upper stratosphere in agreement with satellite observations.

> *p.1, l.17: What is a "serious" decrease? Perhaps "fast" or "large" is more appropriate language?*

After the large stratospheric ozone decrease due to ozone depleting substances (ODS) [...]

> *p.1, l.20: Replace by "[...] Antarctic ozone concentrations started [...]".*

Corrected, also in accordance with suggestion of Referee #1.

As a consequence, stratospheric ozone concentrations over Antarctica started to increase again [...]

> *p.2, l.1: Replace by "[...] increasing ozone levels are more difficult [...]".*

Changed according to the comment of Ref. #1.

Outside of the polar regions, however, differences in ozone recovery are observed depending on altitude and latitude.

> *p.2, l.1: Replace by "[...] to detect and these depend on altitude [...]".*

See previous comment.

> *p.2, l.7: Add "s" to "[...] greenhouse gas increases [...]".*

Adapted to "the increase of greenhouse gases".

> *p.2, l.19-20: Replace by "[...] with no or small drifts [...]".*

[...] Sofieva et al. (2017) used only stable satellite products with no or small drifts.

> *p.2, l.25: Remove "data" from "[...] data steps [...]".*

Corrected.

> *p.2, l.27: Move the "yet" to "[...] has not yet been [...]".*

[...] but it has not yet been applied to ground-based data.

> *p.3, l.21: Replace by "[...] They measure the 142 GHz line where ozone molecules [...]". Currently you have "[..] atmospheric emission [...] molecules emit [...]".*

They measure the 142 GHz line where ozone molecules emit microwave radiation due to rotational transitions.

> *p.3, l.29: Replace by "[...] is the main focus [...]".*

The **Gro**und-based **M**illimetre-wave **O**zone **S**pectrometer (GROMOS) located in Bern, Switzerland, is the main focus of this study (Kämpfer, 1995; Peter, 1997).

> *p.4, l.10: Replace "instead" by "rather than from".*

Corrected.

> *p.4, l.12: "data corrections".*

Corrected.

> *p.4, l.13: "troposphere" is possibly more precise terminology here, rather than "atmosphere"? This then also links better with next phrase which contains "tropospheric humidity".*

Because the stratospheric signal is weak in an opaque and humid troposphere, [...]

> *p.4, l.16: "[...] 3-day moving [...]". Drop "s".*

[...] 3-day moving median window [...]

*p.4, l.18: Replace by "[...] outliers exceeding four times [...]".*

[...] where the instrumental system temperature showed outliers exceeding four times the standard deviation within a 30-day moving median window.

*p.4, l.18: "[...] 30-day moving [...]". Drop "s".*

[...] 30-day moving median window [...]

*p.4, l.34: Move the last phrase "The lidar can only [...] the influence of sunlight." to the first paragraph of Sect. 2.1.2. This to avoid confusion that this information holds for all lidars, not just the one installed at Hohenpeissenberg which is the subject of this paragraph.*

Corrected.

*p.5, l.9: Remove newline, and ensure the phrase "More detailed information [...]" is next to the one ending in "[...] with the GROMOS limits." Since the phrase only contains information for OHP.*

Corrected.

*p.5, l.28: Replace by "[...] 30 km, above which the balloon [...]". The balloon does not usually burst at "30 km", but at "30 km or above".*

Ozonesonde data are limited to altitudes up to $\sim 30\,\text{km}$, above which the balloon usually bursts.

*p.6, l.2: Replace by "[...] Aura satellite, launched in mid 2004, [...]".*

The microwave limb sounder (MLS) on the Aura satellite, launched in mid 2004 [...]

*p.7, l.4: Replace by "The vertical resolution of GROMOS and SOMORA are usually coarser than for the other instruments. [...]".*

The vertical resolution of GROMOS and SOMORA is usually coarser than for the other instruments. When comparing profiles directly with GROMOS profiles, the different vertical resolution of the instruments has to be considered.

*p.7, l.17: Replace by "[...] GROMOS and SOMORA have a higher temporal resolution than the other instruments. [...]".*

Corrected.

> *p.8, l.18: "temporally correlated".*

Such inhomogeneities lead to groups of temporally correlated data [...]

> *p.8, l.22: Remove "that lead to anomalies". Otherwise "[...] reason for inhomogeneities might be [...] issues that lead to anomalies [...]".*

Several sentences have been rephrased.

Inhomogeneous data can originate from changes in the measurement system (e.g. changes in calibration standards or merging of data sets with different instrumental modes), from irregularities in spatial or temporal sampling, or from unknown instrumental issues. [...] We use their method and code in our trend analyses to account for inhomogeneities. The inhomogeneities are in our case anomalies in some months that we identified as described in Sect. 3.1.2.

> *p.9, l.7: Replace "adjusted" by "scaled", because that is what you do.*

Corrected.

> *p.9, l.8: Replace by "[...] the trend fit becomes unity."*

Corrected.

> *p.9, l.20: Some irrelevant information. Replace by "[...] In a second step, we account for biases [...]".*

In a second step, we account for biases in the data subsets where anomalies were detected.

> *p.9, l.27: Drop "have" from "we have found".*

Corrected.

> *p.9, l.31: Remove "and the Aura/MLS trend starts in Jan. 2005", as this is duplicated in the next phrase.*

Corrected.

> *p.9, l.32: Replace by "Aura/MLS cover the shortest trend period, starting in [...]".*

Corrected.

Aura/MLS covers the shortest trend period, starting only in Jan. 2005.

*p.9, l.33: Add "regression" to "regression model output", to avoid any confusion with e.g. CCM's.*

Corrected.

*p.10, l.28: Replace altitude "classes" by "ranges".*

Corrected.

*p.11, l.1: "[...] whereas the GROMOS data shown here [...]".*

Corrected.

*p.12, l.20: Replace by "[...] per decade, i.e. b = 0.1 [...]".*

Corrected.

*p.12, l.25-26: Replace by "The residuals are of order $10^{-6}$ and increase towards the start and end of the time series (Fig. 5(b))."*

Corrected.

*p.13, l.12-13: Rephrase, it is hard to read. "We further found that the trend estimate is closer to its true value when higher uncertainties are chosen [...]".*

Corrected.

*p.14, l.11: Replace by "[...] even though the GROMOS [...]".*

Corrected.

*p.16, l.22: How do lidar data "derive" from satellite data? Did you mean "deviate", perhaps?*

Yes, it was a mistake, it was corrected to "deviate".

*p.17, l.29: Replace by "[...] in the trend analysis [...]".*

Corrected.

*p.26, Fig. 1: Not "stripes" but "lines".*

Corrected.

*p.32, Fig.7: Remove the space before "case II" in the parenthesis on the second line of the caption.*

Corrected.

[revised manuscript text omitted]

[a] Averaged number of profiles per month in the analysed period.

[b] For the trend calculations, data from Jan. 2005 to to Dec. 2017 are used.

**Table 2.** Summary of the artificial ozone time series and the different model parameters used to correct the trend estimation for artificially added anomalies.

| | Parameters in the artificial time series | | | Parameters in the trend programme | | | |
|---|---|---|---|---|---|---|---|
| Case | True trend | Added anomalies | | Monthly uncert.[a] | Uncert. for anomalies[b] | Bias uncert.[c] | Estimated trend |
| A | 0.1 ppm/decade | – | | 0.1 ppm | – | – | 0.100 ± 0.010 ppm/decade |
| B | 0.1 ppm/decade | 5 % | | 0.1 ppm | – | – | 0.133 ± 0.010 ppm/decade |
| C | 0.1 ppm/decade | 5 % | | 0.1 ppm | 0.36 ppm | – | 0.103 ± 0.011 ppm/decade |
| D | 0.1 ppm/decade | 5 % | | 0.1 ppm | – | 5 ppm | 0.100 ± 0.011 ppm/decade |
| E | 0.1 ppm/decade | 5 % | | 0.1 ppm | 0.36 ppm | 5 ppm | 0.100 ± 0.011 ppm/decade |

[a] Uncertainty considered in the diagonal elements of the covariance matrix $\mathbf{S_y}$.

[b] Uncertainty considered in the diagonal elements of $\mathbf{S_y}$ for months with anomalies.

[c] Bias uncertainty considered in the off-diagonal elements of $\mathbf{S_y}$ for months with anomalies, set as a correlated block.

–